# Cutting the costs of coastal protection by integrating vegetation in flood defences

Vincent T. M. van Zelst [1✉], Jasper T. Dijkstra [1], Bregje K. van Wesenbeeck [1,2], Dirk Eilander [1,3], Edward P. Morris[4,5], Hessel C. Winsemius[1,2], Philip J. Ward [3] & Mindert B. de Vries [1]

Exposure to coastal flooding is increasing due to growing population and economic activity. These developments go hand-in-hand with a loss and deterioration of ecosystems. Ironically, these ecosystems can play a buffering role in reducing flood hazard. The ability of ecosystems to contribute to reducing coastal flooding has been emphasized in multiple studies. However, the role of ecosystems in hybrid coastal protection (i.e. a combination of ecosystems and levees) has been poorly quantified at a global scale. Here, we evaluate the use of coastal vegetation, mangroves, and marshes fronting levees to reduce global coastal protection costs, by accounting for wave-vegetation interaction.The research is carried out by combining earth observation data and hydrodynamic modelling. We show that incooperating vegetation in hybrid coastal protection results in more sustainable and financially attractive coastal protection strategies. If vegetated foreshore levee systems were established along populated coastlines susceptible to flooding, the required levee crest height could be considerably reduced. This would result in a reduction of 320 (range: 107-961) billion USD$_{2005}$ Power Purchasing Parity (PPP) in investments, of which 67.5 (range: 22.5- 202) billion USD$_{2005}$ PPP in urban areas for a 1 in 100-year flood protection level.

[1] Deltares, P.O. Box 177, 2600 MH Delft, The Netherlands. [2] Delft University of Technology, Faculty of Civil Engineering and Geosciences, P.O. Box 5048, 2600 GA Delft, The Netherlands. [3] Institute for Environmental Studies(IVM), Vrije Universiteit Amsterdam, 1081 HV Amsterdam, The Netherlands. [4] Instituto Universitario de Investigación Marina (INMAR), University of Cádiz, 11510 Puerto Real, Cádiz, Spain. [5] Cervest, London EC1V 9HX, UK. ✉email: Vincent.vanZelst@deltares.nl

Globally, about 600 million people are at risk of coastal flooding, of which 320 million are in urban areas[1]. Future population growth and urbanization will expose an increasing number of people and amount of assets to coastal flooding[2,3]. Originally, coastal areas are solely protected by natural features. However, coastal areas where these features alone are insufficient, where occupied land is low lying or where people have encroached coastwards are often protected from floods using human-made structures. These structures are also known as 'grey' coastal protection, such as seawalls and levees. Nowadays Nature-based Solutions (NbS) are considered potentially sustainable and cost-effective complements to engineered flood defences[4–6]. For example, foreshores vegetated with salt marshes and mangroves can substantially reduce incoming wave heights[7–10], meaning that levees protecting the hinterland can be lower than 'grey' coastal protection, resulting in reduced initial investment costs and maintenance costs. Besides flood hazard reduction during storm events, marshes and mangroves provide many other ecosystem services such as: carbon storage, habitats for fish and birds, improved water quality and accumulation of sediments[11,12]. These wetland areas are decreasing globally[13,14]. Coastlines are inherently dynamic and are either accreting, eroding or stable[15,16]. In the short term (coming decade), the structural loss of wetlands is likely to result in extra investment costs for coastal infrastructure, and in the long term (this century) it may lead to widespread loss of coastal lands and relocation of millions of people inhabiting coastal areas.

As flood risk is expected to increase in the future as a result of rising extreme sea levels and socioeconomic developments, there is a need to increase coastal resilience accompanied with a strong demand for coastal flood protection measures[17,18]. The role that coastal vegetation can play in reducing coastal flood risk has been quantified in several local, regional, national and global studies[5,19–22]. Only few global studies have been based on process-based wave modelling, and those that have taken this approach focused on mangroves only[23]. While these studies exemplify the role of coastal vegetation as ecosystem services to reduced flood risk, the potential of combining coastal vegetation with traditional flood defence measures so-called hybrid or green/grey protection)—has not been assessed on a global scale. Such an assessment is important because hybrid protection can be very effective. Small- to medium-sized coastal vegetation belts alone cannot prevent inundation[21], while levees without protective coastal vegetation in front require larger dimensions, and hence investments, for the same level of safety compared to hybrid protection. Thus conserving coastal ecosystems and accounting for coastal ecosystem presence fronting 'grey' coastal protection could be cost-effective. Considering hybrid protection on a global scale is unique as it bridges the gap between studies on ecosystem services and flood risk reduction.

A vegetated foreshore in front of a levee reduces wave height and thereby wave run-up and wave overtopping[10,24,25]. Consequently, integration of vegetated foreshores in coastal protection systems, as a supplement to seawalls and levees, allows for lower crest heights[24] and results in hybrid coastal protection systems that are more adaptive to new information and conditions that may emerge, such as sea-level rise (SLR)[25]. In this study we focus on wave-vegetation interaction and as a conservative approach we neglect the effect of coastal vegetation on storm surge levels, because surge reduction is typically only relevant for very extensive coastal vegetation belts[26] and is largely dependent on the local spatial configuration[27]. In addition, coastal ecosystems have the ability to build up vertically due to the accumulation of sediments caused by biophysical feedback mechanisms[28]. Hereby, coastal ecosystems can alter the intertidal elevation profile. This study focusses on the present situation, but (future) changes in intertidal elevation (e.g. due to reduced sediment supply or SLR) will influence propagation of both waves and storm surges[29].

To assess the effect of coastal vegetation on reducing levee crest height, we assumed the presence of a levee at the back of the vegetated foreshore. Global information on the presence of coastal levees is not available. In addition to available global elevation and bathymetric maps more accurate intertidal elevation data (20 m horizontal resolution, 0.52 m RMSE vertical accuracy) are obtained by creating a new global data layer based on time-ensemble average satellite images of the probability of inundation (Methods). Similarly, to acquire high-resolution data on the presence of coastal vegetation, Sentinel-2 A and Landsat-8 images were used to construct a global vegetation map using NDVI values (Methods). This map was combined with existing vegetation maps[14,30–32] to add vegetation type, focusing on marshes and mangroves. As forcing, we used wave heights and periods from a re-analysis of ERA-Interim[33] and extreme water levels from a global tide and surge model[34].

Coast-normal transects were defined for global coastlines ranging between 66° N and −60° S, with alongshore distances of ~1 km. For each transect, a bathymetric profile, vegetation cover and hydrodynamic boundary conditions were determined based on the described data sources. The boundary conditions were derived for nine flood level return periods (2, 5, 10, 25, 50, 100, 250, 500 and 1000 years). We translated offshore wave heights to nearshore conditions and calculated foreshore wave propagation (Methods). Wave damping by vegetation was obtained by comparing wave propagation over the transect with and without vegetation. For both situations, we determined the required levee crest height to prevent flooding, assuming the same landform and a levee at the back of the vegetated foreshore (Supplementary Fig. 1). The difference in crest height between the two situations was used to monetize the effect of coastal vegetation, using unit investment costs of levees that are corrected for differences in construction costs across countries (Methods). We identified populated coastal areas susceptible to flooding based on inundation using flood maps of 1 km resolution and different population density classes (Methods). Finally, a distinction is made between urban and rural areas (Methods).

Here, we show the first global overview of the protective value of hybrid coastal protection, in which coastal vegetation is integrated in vegetated foreshore-levee systems. We assess the potential reduction in costs compare to traditional 'grey' coastal protection. The resulting cost saving represents a reduction in coastal protection investment costs. This also applies to areas where levees already exist, as including the effects of vegetation may imply that costs for future levee heightening, for example due to projected sea-level rise, are not necessary. For areas where no levees currently exist, this method gives an approximation of the costs that could potentially be saved. In the current study we use open source earth observation (EO) data and tidal statistics to produce unprecedented high-resolution global intertidal elevation maps (20 m horizontal resolution, 0.52 m RMSE vertical accuracy) and vegetation maps (10 m resolution) (Methods) to overcome data scarcity in the intertidal zone. In combination with other open source global datasets[14,30–32,35,36], we use these data in a numerical model of wave attenuation to obtain the first process-based global assessment of flood hazard reduction by coastal vegetation in hybrid coastal protection systems. This assessment offers valuable insights in locations where coastal vegetation is of great importance and where hybrid coastal protection systems could be applicable.

## Results

**Coastal vegetation presence.** We find that 18.5% of the global coastline is covered by marsh or mangrove vegetation. This is mostly a fairly narrow (25–250 m) vegetated belt, which covers 6.3% of the global coastline. In addition, 3.9% and 3.3% is

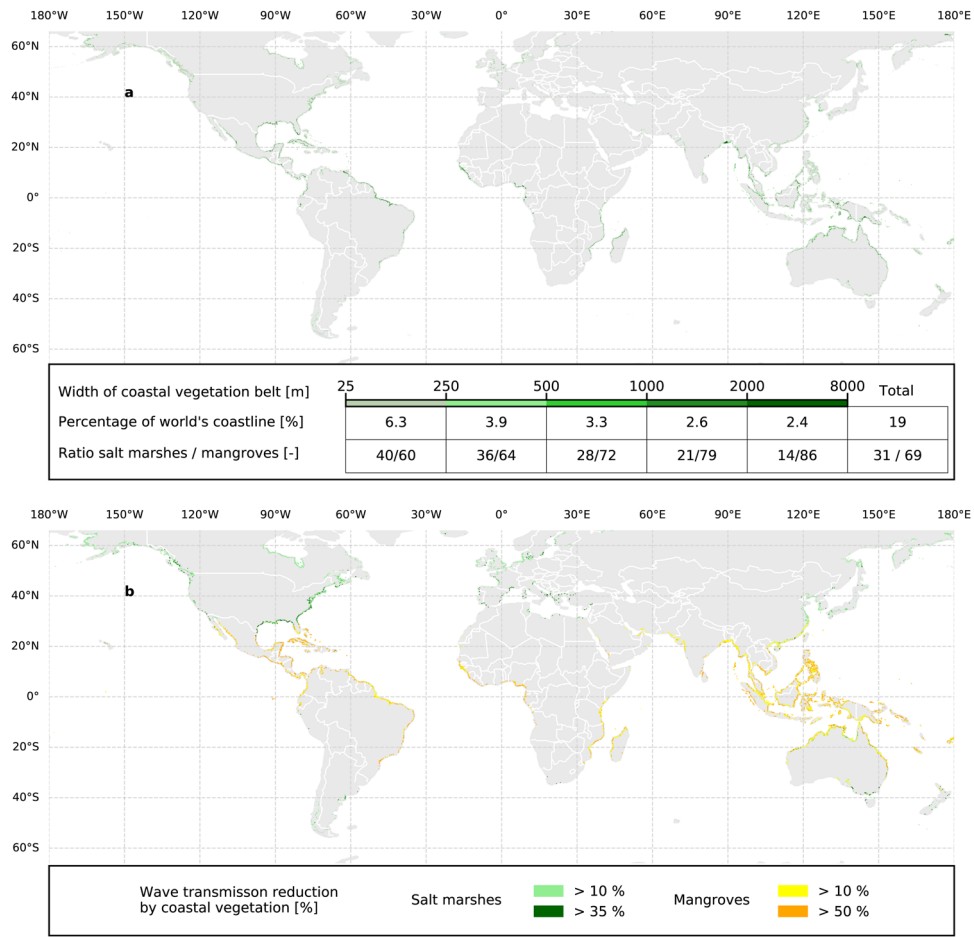

**Fig. 1 Global distribution of coastal vegetation and resulting wave transmission reduction. a** Global distribution of coastal vegetation belt width in metres. **b** Reduced wave transmission solely due to wave-vegetation interaction based on storm conditions with a return period of 100 years along global coastlines susceptible to flooding. Map is created with Python 3.8.10 (https://python.org) using Cartopy (v0.18.0. Met Office UK. https://pypi.python.org/pypi/Cartopy/0.18.0), GeoPandas v0.8.1 (https://geopandas.org) and Matplotlib v3.3.4[97].

covered, respectively, with a vegetation belt of 250–500 m and 500–1000 m. Furthermore, a vegetation belt exceeding 1000 m width covers 5% of the global coastline (Fig. 1a). Wide vegetation belts occur near the equator, where extensive mangrove forests still exist. Of the world's coastline, 17% is populated (population density greater than 1 per km²) and susceptible to coastal flooding (hereafter referred to as the susceptible coastlines). Of the rural and urban populated coastlines susceptible to flooding, 25% and 18.5% are vegetated, respectively (Supplementary Fig. 3). Wider vegetation belts generally occur along rural coastlines. On average, the width of mangrove belts in populated and flood susceptible locations is 858 m in urban areas and 1233 m in rural areas. For marshes, the average width of the vegetation belt in populated and flood susceptible locations is 483 m (urban) and 613 m (rural) (Supplementary Fig. 3).

**Wave attenuation by coastal vegetation**. We find that for 11.5% of the global coastline, vegetated foreshores can reduce incoming wave heights by more than 25%, solely due to wave-vegetation interaction (Fig. 1b). This figure is calculated by comparing wave heights for a return period of 100 years between bare and vegetated foreshores. Despite the large mean width of the vegetation belt in rural areas, the difference in wave reduction between rural and urban areas is limited. This is mainly due to the non-linear relationship between wave attenuation and vegetation width[37] (Supplementary Fig. 3).

**Impact on required levee crest heights**. For 27.6% of populated susceptible coastlines, the current presence of coastal vegetation allows for lower levee crest heights while maintaining the same protection standard. The required crest heights for a 100-year protection standard are reduced by 25–50 cm for 5.8% of the susceptible coastline, 50–75 cm for 9.5% and by more than 75 cm for 12.3%. Across all vegetated susceptible coastlines, the mean crest height reduction for a return period of 100 years is 96 cm. For return periods of 2 and 1000 years, the mean crest height reductions are 82 and 104 cm, respectively (Supplementary Fig. 5a). For 22.1% of the susceptible coastlines, the current presence of coastal vegetation allows for a reduction in the required crest height equal to or greater than the projected sea-level rise of 0.49 m (RCP4.5) by the end of the 21st century[38].

**Results on country level**. If levees with a 100-year protection standard were constructed along all populated susceptible coastlines, the potential reduction in costs resulting from the presence of current foreshore vegetation presence is 320.2 billion USD (all monetary values in USD₂₀₀₅ PPP), of which 67.5 billion USD is in urban areas (Fig. 2a). The ten countries with the highest reductions in cost are shown in Fig. 2a. These ten countries all have relatively long coastlines and account for over 65% of the total potential reduction in costs, with a combined potential of 208.6 billion USD. The average levee crest height reduction per kilometre for susceptible coastlines is 0.24 m km⁻¹ globally, with the

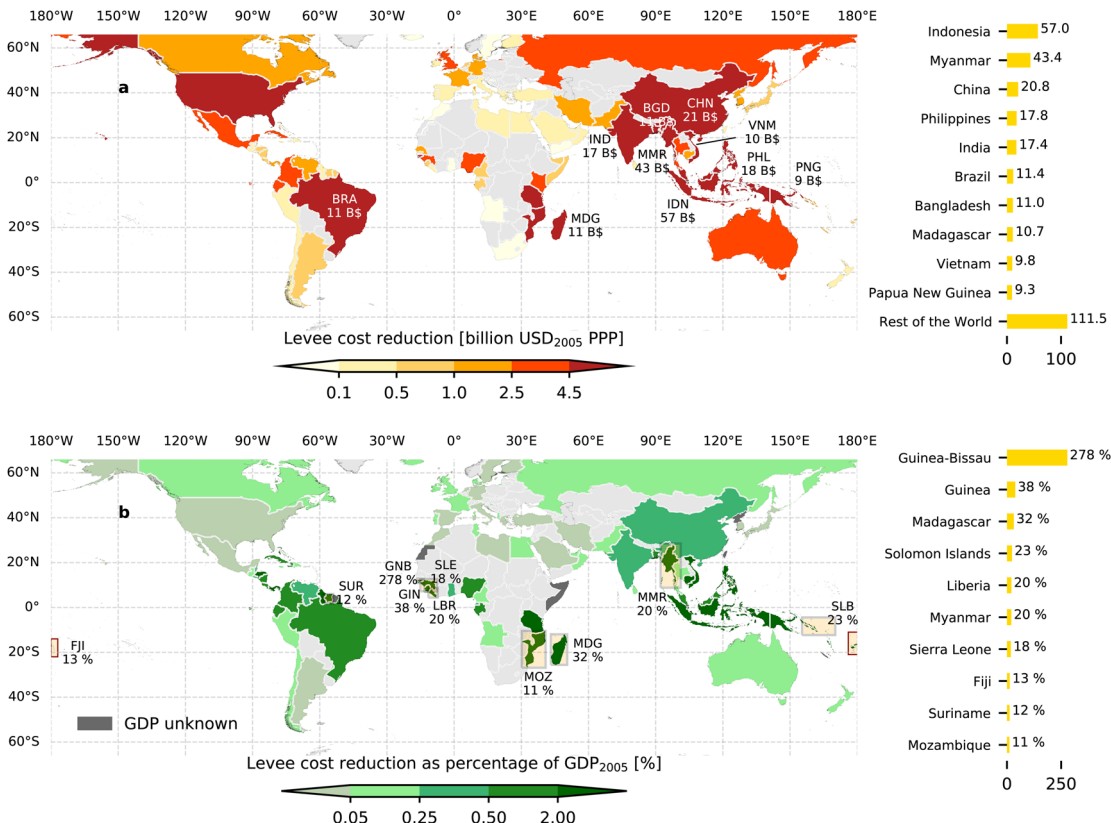

**Fig. 2 Reduced coastal protection costs with annotations for the top 10 countries. a** Potential levee cost reduction by coastal vegetation per country in billion USD$_{2005}$ PPP. **b** Potential levee cost reduction by coastal vegetation in urban areas per country as percentage of GDP$_{2005}$. Values in these maps apply to a 100-year protection standard and a 'Medium' levee unit cost scenario. Non-benefiting countries are indicated with light grey colour. Map is created with Python 3.8.10 (https://python.org) using Cartopy (v0.18.0. Met Office UK. https://pypi.python.org/pypi/Cartopy/0.18.0), GeoPandas v0.8.1 (https://geopandas.org), Matplotlib v3.3.4[97] and GADM v2 (https://gadm.org) administrative boundaries.

highest average value in Guinea-Bissau (0.77 m km$^{-1}$) (Supplementary Fig. 4). Expressing the potential saved investment costs as a percentage of the country's gross domestic product (GDP) highlights the importance of current coastal vegetation for Small Island Developing States (SIDS) (Fig. 2b). For many of these countries, 'grey' coastal protection along their full coastline is not socially and financially feasible[39]. Hence, coastal vegetation as part of hybrid coastal protection offers opportunities at these places.

The majority of the coastlines along which we find potential to implement hybrid coastal protection have a low to medium exposed population (1–100 people km$^{-2}$) (Fig. 3). Eight out of the top ten countries with the potential to reduce coastal protection costs by applying hybrid coastal protection as Nature-based Solutions in higher populated areas (>100 people km$^{-2}$) can be found in Asia (Philippines, China, Indonesia, India, Vietnam, Myanmar), Africa (Madagascar, Nigeria, Guinea) and Europe (United Kingdom).

On average the width of the vegetation belt decreases with increasing population density (Fig. 3), although this trend is not statistically significant given the large error margins. This can be related to the fact that vegetation extent has been decreasing for decades as a result of conversion of green belts for purposes such as agriculture or urban land use[40]. As a result, the potential wave height reduction by mangroves and salt marshes decreases for vegetation belts with lower widths. The hydrodynamic interaction between waves and vegetation depends on physical parameters such as the bathymetry profile, surge level, wave height, vegetation width and height. While these parameters are important for the correct calculation of wave height reduction,

uncertainty in the levee cost reduction results are mainly determined by uncertainty in levee construction costs (Supplementary Fig. 10). To account for this uncertainty we applied three different standard levee unit cost scenarios (low, mid, high) in line with previous research[41,42]. The difference between the high and low levee cost scenarios is far greater than the difference between a low standard of protection (RP5) and a high standard of protection (RP1000) (Fig. 3).

## Discussion

This global study highlights where including coastal vegetation in hybrid coastal protection schemes is applicable and provides insight in the total potential cost reduction of these Nature-based Solutions. Existing coastal vegetation can contribute substantially to cost-effective flood risk reduction strategies for countries along a substantial part of the global coastlines. The ability of coastal vegetation to reduce wave impact is vital for countries with long rural coastlines where conventional protection by hard infrastructure alone might be economically unfeasible (Fig. 3e). In addition, several small island developing states receive major benefits from the wave-reducing abilities of coastal vegetation (Fig. 2b). Hybrid coastal protection schemes combining coastal vegetation and levees or seawalls, have large potential in countries such as China, United Kingdom and Indonesia, where coastal vegetation helps to protect the highest number of people and amount of assets. The use of hard flood risk protection structures is widespread around the globe. With SLR and growing population in vulnerable areas, future economic benefits and the need for protective infrastructure are expected to be larger[17,28,43,44].

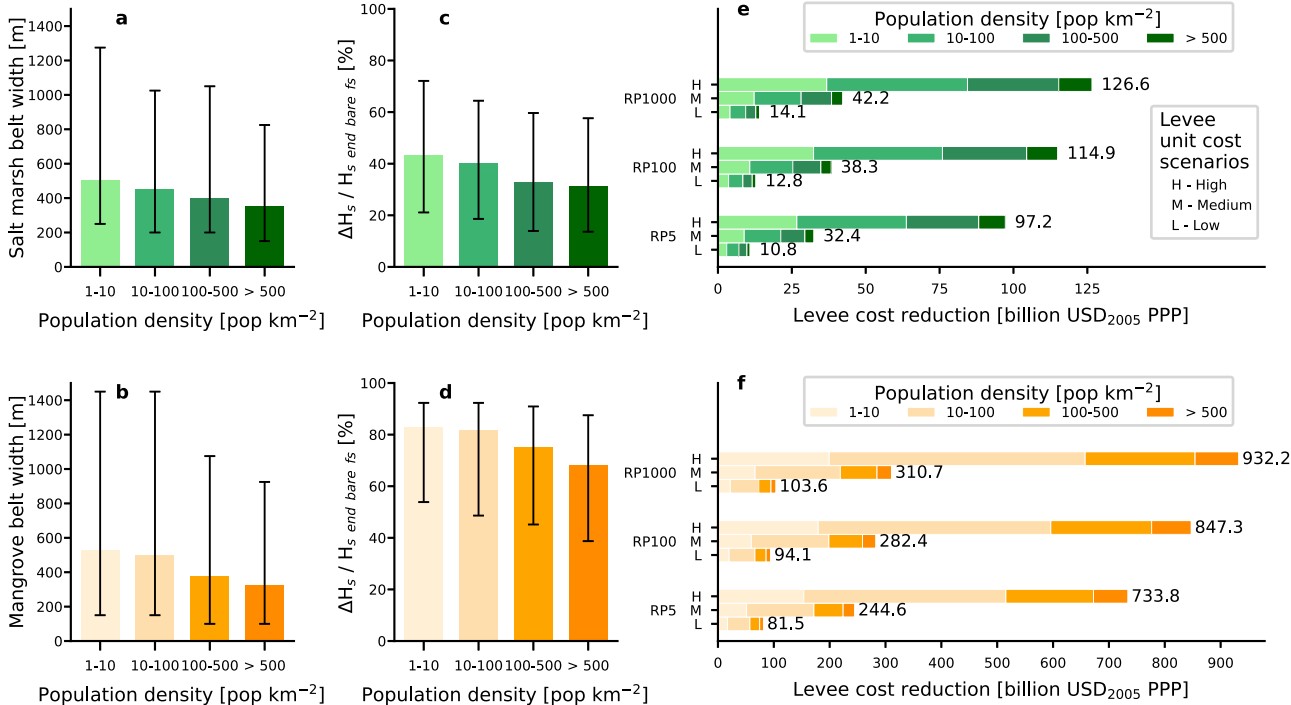

**Fig. 3 Role of salt marshes and mangroves for areas characterized by various population densities.** Salt marsh (**a**) and mangrove (**b**) width of vegetation belt for four classes of population density. Wave attenuation on foreshores vegetated by salt marshes (**c**) and mangroves (**d**) in comparison to bare foreshores. The bars represent the upper 85%-ile and the 15%-ile. Levee cost reduction for three levee unit cost scenarios, three standards of protection and four classes of population density for both salt marshes (**e**) and mangroves (**f**).

Results of the current study should be considered as a warning not to remove coastal vegetation and to regulate coastal developments strongly, both along rural and urban coastlines. Mangroves and salt marshes play an important role in keeping these coastlines stable and could considerably lower the costs of coastal protection infrastructure. Moreover, next to the benefits that coastal vegetation can provide to coastal flood protection, other benefits of coastal vegetation include positive impacts of tourism[45], regulating fish and shellfish stocks and playing a role in the sequestration of carbon[12].

Global mapping studies that combine models and different data sources inevitably accumulate uncertainties. For example, the limited spatial resolution affects the accuracy of the metrological forcing data used from ERA-I, the Global Tide and Surge Model (GTSM), and the resulting estimates of extreme waves and water levels. ERA-I waves have a bias between 0 and -0.2 metres, which results in an expected error (scatter index) between 16 and 20%[33]. Extreme water levels show an average relative error in the range of 11–14%[34]. Further, the newly developed intertidal elevation data constructed using EO data and tidal epoch data has a vertical accuracy of 0.52 m RMSE (Supplementary Fig. 7). Note that to be conservative we did assume the same landform for the simulations with and without vegetation. For this study, each data layer was validated with local data or local models, and a qualitative reliability analysis is performed (Supplementary Figs. 6, 7, 8, 9). We performed a sensitivity analysis (Methods) to identify the factors that have the largest influence on the coastal reduction costs (Supplementary Fig. 10). Despite the large uncertainty in the hydrodynamic data and topographic data used, our analysis shows that the levee costs reduction potential is influenced most by assumptions on the required coastal protection needs (people exposed) and associated levee construction costs (Fig. 3, Supplementary Fig. 10). This implies that the current study can be improved by further limiting the uncertainty in the levee costs and requirements, activities that are typically undertaken in a

local assessment. Deviations in the width of the vegetation belt play a minor role in comparison to the aforementioned variables, because of the non-linear relationship between wave height reduction and vegetation width[37]. Global assessments have their limitations, and mismatches between model abilities and user expectations should be prevented[46]. The current study provides a global overview and is meant to stimulate detailed local assessments in areas where it matters most. We stress that local assessments are required for the correct design and implementation of Nature-based Solutions.

The percentage of the susceptible populated coastline benefitting differs for the various protection standards (Supplementary Fig 5), with 19.2% and 23.2% for a 2 and 1000 years protection standard respectively using a critical wave overtopping rate of $1\,l\,s^{-1}\,m^{-1}$. Here, benefitting is defined as having a reduction in required levee crest height of at least 50 cm due the presence of coastal vegetation. This spread (19.2–23.2%) is smaller in comparison to the outcomes for various critical wave overtopping rates. For rates of 0.1, 1.0 and $10\,l\,s^{-1}\,m^{-1}$ the percentage of coastlines benefitting is 25.6%, 21.8% and 11.1%, respectively, for a 1 in 100-year protection standard. These results point out that the reduction of the required crest height is largest for simple earthen levees, as the applicable critical wave overtopping rate depends mainly on the quality of the levee[47]. Consequently, reduced levee heights in countries with more strict levee quality are smaller, as the reduction in required levee height due to the presence of coastal vegetation is less. However, the associated cost savings of high quality and low quality levees might be comparable. A smaller levee crest height reduction for high quality levees might be balanced out, because high quality levees may actually be more expensive.

For this study the effects of vegetation on storm surges were not included. Although several studies mention the positive effects of marshes on surge reduction[24,29], other studies emphasize that surge reduction depends strongly on storm duration and intensity, and that effects are only significant for mild and short

storms[48] and that surge levels of severe storms are comparable for bare tidal flats versus marshes with widths below 1200 m[6]. This suggests that surge propagation is context specific and influenced by local features, specifically nearshore geomorphological configuration. Conservatively, we assumed that to reduce surges, vegetated foreshores of multiple kilometres are required. Such large wetlands only occur on a few deltas around the world, with only 4 out of 11 providing effective protection[26]. Most of the vegetation belts along populated susceptible coastlines have an insufficient width (median belt width of 525 m) to have substantial effects on surge reduction. Vegetation belts with widths exceeding 2 km only exist for 3.0% of the vegetated transects in urban areas susceptible to flooding.

The resilience of coastal vegetation under severe conditions is uncertain[49]. Damage reports following severe events, such as hurricane Haiyan, show destruction of planted mangroves that were in the path of the typhoon[50]. Uprooting and shear strength influence the ability of coastal vegetation to reduce wave heights during storm conditions, but are not captured by the current model. For this study we determined coastal vegetation belt widths, but we used spatially constant (conservative) vegetation characteristics (Methods). For salt marshes, we used a winter state as found in NW Europe, and for mangroves we used characteristics of young fringing pioneer mangroves (Methods). In reality salt marshes and mangroves are complex habitats with considerable spatial variation due to creeks, differing vegetation cover and sediment characteristics. Performing the current analysis with spatially varying vegetation characteristics (such as mangrove canopy height[51] and root density), based on new EO data techniques, would be an interesting topic for future study. In the current study, we tried to make a global assessment while approaching reality as much as possible by focusing on the main factors of relevance and reducing uncertainty by developing improved global data layers and by validating results. Field evidence to support the role of wave attenuation by vegetation is extremely limited for more extreme waves and surges. Therefore, implementation of formulations and results of field studies during non-storm conditions, scaled lab studies and results of large-scale salt marsh flume tests[10] in (numerical) models are to date the best way to gain insights on the buffering role of coastal vegetation for more extreme conditions. Future large-scale mangrove flume experiments may provide a better evidence base and improved understanding of the role of mangroves on wave attenuation during extreme storms.

Our study focused on storm conditions, which have a low probability of occurrence and last for a short period. Sea-level rise (SLR), on the other hand, is a slower and long-term process with far-reaching consequences. The effect of SLR on intertidal areas depends on complex hydrodynamic interactions between tides, waves, wind, fresh water run-off (for estuarine wetlands) and sediment supply. These complex interactions hamper the long-term modelling and prediction of the effects of accelerated SLR[52]. On intertidal mudflats, wave energy is re-distributed and dissipated, which limits wave action on coastal vegetation present at the landward limits of the intertidal zone. The decay of intertidal flats will therefore put more stress on existing salt marshes and mangroves[53]. In addition, mechanisms such as coastal squeeze will determine the future state of coastal ecosystems and their coastal vegetation belts locally[54]. For vegetated areas the situation is even more complex, as also root growth, compaction and subsurface faunal processes play a role[55]. Despite these difficulties, multiple studies[28,56–58] show the ability of vegetated foreshores to accumulate sediments and thereby keep up with SLR, but also underline the presence of tipping points. Studies indicate that once the rate of SLR exceeds a context dependent critical threshold, vegetated foreshores cannot keep up with SLR resulting in die-off of the lower vegetated part first[57,59,60]. Assessing the potential loss and degradation of coastal ecosystems due to the effects of SLR and coastal squeeze and the resulting effect of the partial loss of vegetation's wave buffering capacity on coastal protection costs would be an interesting field for future study.

Global analyses involve inevitable assumptions and shortcuts. In this study a replacement costs method was used to express the benefits of applying hybrid coastal protection over purely 'grey' coastal protection, hereby assuming that levees would be built along global coastlines with coastal protection needs. However, the answer to the question to protect or not to protect is not easily answered. For example, the protection level against flooding differs per country and can be supported by an analysis of investment costs versus avoided damages[61]. The local evaluation of different flood risk reduction strategies should be done case-by-case. Flood risk could depend on local deviations in topography that are not captured in this assessment that uses global data and 1 km spaced coast-normal transects. In addition, we considered solely locations where vegetation is currently present. Rehabilitation attempts of vegetated foreshores should consider eco-morphodynamic requirements[62] for a more sustainable use of ecosystems, as vegetated foreshores are part of the ecosystem and cannot be considered as isolated structures. Also, levee location is assumed directly behind the vegetated foreshores, but the exact levee position is best determined in a local assessment taking into account factors including the presence of infrastructure (e.g. houses and roads), subsidence[63] and the resilience of the vegetated foreshores[64] (including for example wave reflection from the levee that can hinder geomorphological development[65]). With SLR the latter becomes more important, as levee construction can limit landward migration of vegetation in the long run[66].

Considering the current climate and biodiversity crisis, integration of coastal ecosystems in coastal protection schemes is essential. Currently, levees and seawalls are also regularly found along coastlines with little exposure or hazard (Fig. 4). In these cases, their negative impact by disturbing natural flows of water and sediment and thereby reducing coastal resilience may not outweigh their positive impacts in terms of protecting people or infrastructure. In addition, they are often constructed seaward of coastal vegetation, which means that (a) the survival of vegetation itself is threatened; and (b) the wave-reducing potential of the vegetation is not used. Moreover, to limit future coastal ecosystem loss and to work towards cost-effective and sustainable coastal protection schemes, hard infrastructure should be used with care and vegetated foreshores should be formally incorporated in coastal protection schemes.

Our study shows that integrating present coastal vegetation into infrastructural interventions could result in considerable cost savings. Ongoing removal of coastal vegetation will increase wave forces and is thereby likely to increase erosion, which may result in considerable land loss[11,60,67]. Allowing the removal of coastal vegetation for short-term individual or industrial profits might result in large community costs through resulting land loss and by raising the costs for future coastal protection infrastructure. Formal integration of vegetated foreshores with infrastructure design and sustainable management of coastal ecosystems is a cost-effective and sustainable route towards maintaining many coastlines around the world.

## Methods

**Coastline segments**. For reasons of data availability and socioeconomic relevance, the analysis was limited to latitudes between 66° N and −60° S. In this area of interest, the world was divided in 1 arcmin (~2 km) grid cells. To define a logical position for the establishment of an efficient levee, the coastline location was derived from the OpenStreetMap[68], moved 100 m land inward and smoothed. For

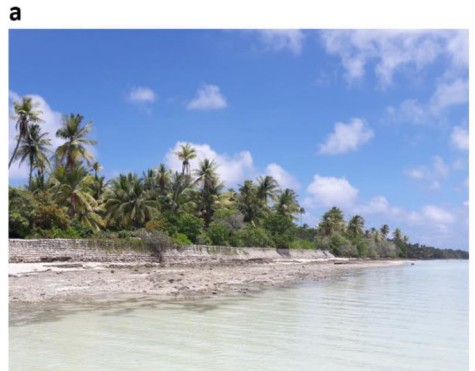
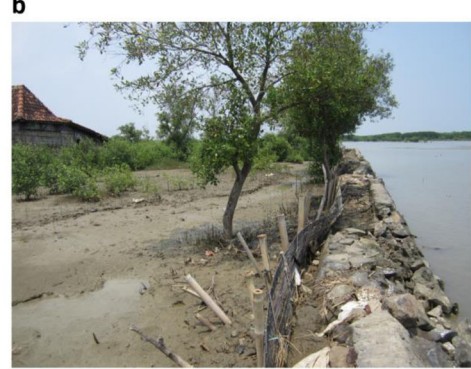

**Fig. 4 Examples hard structures. a** Failing seawall protecting a road in front of coastal vegetation on an outer island of Kiribati (B. van Wesenbeeck). **b** Revetment built in front of a mangrove forest in Demak (Java, Indonesia) (B. van Wesenbeeck).

every cell containing a coastline segment, coastline length and a coast-normal transect were derived at the center of segments resulting in 495.361 transects that are on average 1.1 km apart. Bootstrapping revealed that transect distances up to 2 km give very similar results. All transects stretch 4 km seaward and 4 km inland to fully capture most foreshores.

**Elevation data**. A global intertidal bathymetry/elevation dataset from high-resolution EO data (USGS Landsat and Copernicus Sentinel-2), the Foreshore Assessment using Space Technology (FAST) intertidal elevation map[69], was produced to compliment commonly used global data products with low resolution and higher inaccuracy in intertidal zones. Global coastlines were divided over 25000 tiles of each $40 \times 40$ km². For these tiles, all available images were collected for the period between 1997 and 2017. Surface water was identified, using normalized difference spectral indices (NDSI, here SWIR1 and Green band) for all images (median of 317 images per tile) covering various tidal conditions, and the per pixel mean calculated to derive time-ensemble average (TEA) NDSI images. We developed a new technique to transform TEA images to intertidal elevation independently of in situ calibration data. TEA-NDSI images were normalized by the spatially averaged NDSI values of regions identified (using global elevation datasets) as land and water, respectively. This resulted in a single image per tile that represented the inundation probability for each pixel in the intertidal zone. The inundation probability represents the long-term average tidal inundation, because it was derived from a collection of images that span a time period similar to the tidal epoch (period of 19 years). Pixels having a probability of 1 represent permanent water, and have elevations less than or equal to the lowest astronomical tide (LAT), whereas land ($p = 0$) represents elevations higher than or equal to the highest astronomical tide (HAT). By deduction, $p = 0.5$ is equivalent to local mean sea level (LMSL). Tidal statistics from the global tide model FES2012 were used to couple the derived inundation probability to an elevation. The main source of bed level data originates from this map and has a 20 m horizontal resolution and typically a 30–50 cm vertical accuracy (RMSE = 0.52 m, MAE 0.42 m, as assessed at a number of sites with high quality elevation data (Supplementary Fig. 7)). Bathymetry data (GEBCO[35]; 30 arc-second horizontally, tens of metres vertically) and topography data (MERIT[36]; 3 arc-seconds, 2 m vertically) were merged to create a continuous bathymetry-elevation map by changing the vertical datum of MERIT from EGM96 to MSL by assuming 0 m +MSL at the OSM coastline. Global bathymetry datasets (e.g. GEBCO) and elevation datasets (e.g. SRTM and MERIT) lack accuracy (especially nearshore), but are commonly used[17,18,23,34]. The final bed level was constructed using FAST intertidal data where sufficient valid data points were available, complemented by the merged GEBCO-MERIT data where these points were lacking.

**Vegetation extent**. The FAST coastal vegetation map[69] was based on Landsat-8 and Sentinel-2 satellite images collected between 2013 and 2017. The map provides actual vegetation presence at 10 m resolution. Vegetation presence was obtained by applying an individual NDVI threshold per tile, with a total of 25,000 tiles, based on the yearly NDVI average and NDVI amplitude. The FAST coastal vegetation map is validated based on NDVI comparison with local measurements taken at Zuidgors, The Netherlands (R² = 0.92) (Supplementary Fig. 8). If vegetation was present, the vegetation type was determined by global salt marsh[32] and mangrove[14] maps, complemented with Corine Land Cover[30] (CLC, Europe only) and Glob-Cover v2.2[31] maps when there is no coverage. Determining global coastal vegetation extent is difficult and affected by eutrophication in coastal environments. This behaviour is observed on the coast along the Persian Gulf and the Red Sea. To improve accuracy only vegetated transects identified by the global salt marsh[32] and mangrove[14] map and confirmed by the FAST coastal vegetation map are included for these areas. Moreover, vegetated transects with a green belt width smaller than 250 m identified by GlobCover are excluded from the study for accuracy reasons (Supplementary Fig. 8). To avoid mixed vegetation types, the vegetation type was

determined by the most dominant type. The vegetation width constituted of the sum of vegetated grid cells between the start and the end of the vegetated zone.

**Water level and wave data**. The design water levels were based on a combination of tide and storm surge for the selected probability of occurrence (return periods 2, 5, 10, 25, 50, 100 default, 250, 500, 1000 years) and came from the GTSR dataset[34]. SLR and subsidence were not taken into account because this study focuses on the present situation. Moreover, quantifying the future role of vegetated foreshores would not only require SLR scenarios but also an insight in the development of wetlands over time, which is strongly determined by local conditions such as sediment supply[56,57,60]. Offshore wave conditions were obtained from ERA-Interim[33] re-analysis, based on data from 1979 till 2017 and reprojected to Dynamic Interactive Vulnerability Assessment (DIVA)[70] points. Next, the Peak Over Threshold method was applied to construct representative values for the significant offshore wave height, $H_s$ and the peak wave period $T_p$ for all the return periods. The nearshore wave height was limited by the local water depth at the start of the (vegetated) foreshore using a breaker criterion (gamma = 0.55). This is a fairly low value considering the range of values cited in literature[71] leading to conservative wave attenuation by vegetation results. Wave-bottom interactions in the sub-tidal zone and processes such as refraction and diffraction are not explicitly simulated. The conservative breaker criterion is chosen to implicitly account for these processes in a conservative manner. The wave period remained unchanged and the wave direction was assumed coast normal and wave growth along the transect due to wind effects was excluded. However, for the current study a more sophisticated approach to account for longshore wave variability based on topography was considered infeasible at the global scale and considered to yield limited outcome looking at the uncertainty in socioeconomic factors. The average $H_{s,offshore} = 4.6$ m (std = 2.0 m) and the average $H_{s,startforeshore} = 0.7$ m (std = 0.7 m).

**Profile construction**. The 8 kilometre coast-normal transects consisted of 321 gridpoints, thus a horizontal grid resolution of 25 m. We used four different methods: Foreshore method 1 (based on the FAST intertidal elevation map), Foreshore method 2–4 (based on MERIT-GEBCO). The properties of the FAST intertidal elevation map, MERIT and GEBCO are described under the header 'Elevation data'. Foreshore method 1 produced the most accurate profiles and foreshore method 4 the least accurate profiles. The profile construction steps are described hereafter. Validity checks were performed to identify false indications of intertidal area in the FAST intertidal elevation map. Individual data points were marked invalid and removed in case: (1) MERIT points were situated above the surge level with a return period of 2 years, while data from the intertidal map indicated a lower elevation. (2) Data from the FAST intertidal map was situated at open sea. (3) Data from the FAST intertidal map along the transect dropped below a minimum range threshold of 10 cm. A fourth check was performed based on the continuity of the data. Data from the FAST intertidal map contain discontinuities along the profile. These continuities exist on pixel level due to the use of the modified normalized difference water index and in some instances cloud coverage was preventing full coverage. Lastly, discontinuities arise due to the presence of (high elevated) tidal flats and banks in coastal areas. (4) Data length was defined as the length of continuous data points along the transect. If the data length of a patch decreased below a threshold of 100 m, the points were marked invalid. Gaps between valid data patches were filled using linear interpolation if the gap was smaller than 250 m. Eventually, one, none or multiple valid data patches were found along the transects. See Supplementary Fig. 2 for example transects.

Global coastline shapes range from straight sandy coastal stretches to complex coastlines often found in estuaries. With a transect length of 8 km, the start and the end of the transects could both be situated on land, hampering an unambiguous identification of the foreshore of interest. We designed the algorithm such that the last foreshore was selected. For profiles using data from the FAST intertidal map (foreshore method 1, 50.9% of populated susceptible coastlines), the last valid patch

corresponds to the last foreshore. The inclusion of tidal flats as part of the foreshore was determined based on the gap length. In case no (sufficient, thus not satisfying the minimum data length criterion of 100 m) valid data was available from the FAST intertidal map based on the four described checks, the profile was based on a merged GEBCO-MERIT set (methods 2, 3 and 4), respectively, 46.1%, 3.0% and 0.01%. For the second method, data points were selected between a minimum threshold of $-2$ m MSL and a maximum threshold equal to the surge level with a return period of 2 years. Next, for the selected points the direction of the slope was determined by comparing elevation between the data point concerned and the next data point. This resulted in patches of upward sloping sets of data points between the minimum and maximum threshold. Similar to foreshore method 1, the validity of the patches was checked using data length, gap length and the corresponding thresholds of 100 m and 250 m. The start and the end of the foreshore were determined by the first and last valid point of the last patch. Foreshore method 3 was used if not sufficient foreshore data were available to satisfy the minimum data length threshold (100 m). In these cases, the start of the foreshore was defined as the first upcrossing intersection with $-2$ m MSL along the transect. The end of the foreshore corresponded to the intersection between the elevation profile and the governing surge level with a return period of 2 years. Foreshore method 4 was used if no start and or end of the foreshore could be found. In this case the start and/or end point of the foreshore corresponded to the first and last data point, respectively.

In some cases, elevation for the end of the foreshore was missing due to several reasons. First, the upper part of the intertidal zone was sometimes missing from the FAST intertidal map, due to low frequency of inundation of the upper intertidal zone or cloud cover. Second, bed elevation in mangrove belts was hard to define based on satellite imagery, as the canopy is detected as the earth surface. These uncertainties were counteracted by consulting the mangrove and salt marsh maps. If vegetation was present in one of these maps, the derived foreshore was extended until the end of the vegetated zone. An elevation equal to the surge level with a return period of 2 years was chosen as elevation for extended foreshore points with an elevation exceeding this surge level.

**Vegetation parameters**. As deducting the type and size of mangrove trees and salt marshes from EO data at global scale is not possible (yet), the current modelling approach relies on field and literature observations. For the scope of this research the properties of the mangrove trees occurring at the seaward side of the mangrove belt are the most relevant. To avoid overestimation of wave attenuation in young mangrove forests, the mangrove dimensions are chosen such to be representative for young fringing pioneering mangroves up to a height of 3 m that are practically vertically uniform compared to mature trees. The modelling approach uses four parameters to represent vegetation: height, diameter, number of stems and drag coefficient. The exact characteristics are based on observations in literature[8,9,72–76] ($N = 30$ m$^{-2}$, $d = 35$ mm, $h = 3.0$ m).

High quality observations on wave attenuation by mangroves under storm conditions do not exist. For the drag coefficient the theoretical value, 1, of a rigid cylinder is chosen, because mangrove trunks can be considered rigid. For salt marshes a winter state representative as found in NW Europe is chosen. The values are defined based on FAST field tests (Romania, UK, Spain and the Netherlands) and literature[10,24,77,78] ($N = 1225$ m$^{-2}$, $d = 1.25$ mm, $h = 0.30$ m). A drag coefficient ($C_D$) of 0.19 is chosen, which is the lower limit found during large-scale flume tests[10]. The drag coefficient depends on biophysical characters as well hydrodynamics. The drag coefficient represents drag due to skin friction and pressure differences, but also effects like swaying motion of stems[24]. The 1D modelling approach takes into account gaps in vegetation cover, e.g. due to the presence of channels. Zonation of vegetation types is not implemented, because this level of detail is insignificant in relation to the inaccuracies induced by the use of global datasets.

**Wave attenuation model**. To determine wave attenuation along the foreshore transects and the resulting significant wave heights relevant for the flood defence on a transect, we used a lookup-table approach. The lookup table was generated by combining 668,304 model output values for different combinations of foreshore slopes, vegetation covers and hydrodynamic conditions. The table contained wave heights modelled by XBeach[79] in surfbeat mode (a nearshore numerical wave model that accounts for the presence of vegetation) at regular intervals along a steady slope, both with and without vegetation. XBeach uses for wave-vegetation interaction the rigid cylinder[80] approach and includes an energy sink term to the wave energy balance to implement wave dampening[81]. We used conservative vegetation characteristics, winter state salt marshes and young pioneering mangroves. We characterized foreshores by their width and slope. The foreshore profile was the same for simulations with and without vegetation. The foreshore width was determined by calculating the distance between the start and the end of the foreshore. The slope was estimated using a linear regression. This approach has two advantages over detailed modelling of wave attenuation over all transects: it is much quicker, allowing for iterative improvements of the workflow and it does not suggest the precision one would expect from detailed models but cannot be delivered with global data. Average $H_{s,endforeshore,noveg} = 0.6$ m (std $= 0.5$ m) and $H_{s, endforeshore,veg} = 0.3$ m (std $= 0.4$ m).

**Coastline susceptible to flooding, urban and rural extents and population density**. To assess the need for coastal flood defences, we made a distinction between areas susceptible to coastal flooding and higher, non-susceptible areas. We determined susceptible areas based on possible inundation using coastal flood maps of 1 km resolution for a 1/1000 year surge level. These maps were created with a global geographic information system (GIS) based inundation model that is forced with a spatially varying sea level, accounting for attenuation of the water level due to land surface roughness[82]. A method that is more sophisticated compared to a simple 'bathtub' inundation method. Topographic features, as visible in MERIT, protecting the land from flooding are considered. To classify coastlines as urban or rural a distinction was made based on gridded population from the LandScan database[83] using the 2UP model[84]. A transect is characterized 'urban' if it intersects at least one cell with an urban population with a minimum of 1. Populated coasts have been identified by assigning the population density of the population susceptible to flooding in the proximity of the transects. We used WorldPop2017[85] population data and assigned population to the transects using a buffer of 15 kilometre radius. The population density is the division of the assigned population and the total area of the assigned cells. This procedure is repeated for buffer radius of 5, 10 and 20 km, giving fairly comparable outcomes. Following this approach we found a ratio between rural and urban transects of 73/27.

**Levee crest heights**. The empirical EuroTop formulations[47] gave the required levee heights with respect to water levels and wave heights, assuming the presence of a levee at the end of the vegetated foreshore. We hereby neglected the position and characteristics of levees present in the current situation, as no global dataset of coastal protection structures exists. The assumed levee had a standard 1:3 levee profile without berms and an allowed overtopping discharge of $1 \, \mathrm{l \, s^{-1} \, m^{-1}}$. These parameters are representative for simple, low-cost levees in developing countries but conservative for well-constructed and maintained levees. Consequently, savings on levee heights in countries with strict protection standards are overestimated, as reduction in required levee height due to vegetation presence is likely less than predicted here. However, this may be balanced out by the fact that we calculated with an average national construction cost per kilometre and levees applying to stricter protection standards may actually be more expensive (Supplementary Fig. 5).

**Costs for levee construction and crest height reduction**. The calculated levee crest height reductions were monetized using a levee unit price per kilometre length per metre heightening. We used an unit investment costs of levees (metre heightening per kilometre length) of USD 7.0 million[42]. This estimate represents an average of construction costs in the USA and the Netherlands stated in several studies[86–89]. It pertains to all investments costs, including ground work, construction, engineering costs, property or land acquisition, environmental compensation, and project management. Investment costs per metre heightening are well described by a linear function without intercept[90]. They concluded that for large-scale studies it is sufficient to assume linear costs for each metre of heightening, including the initial costs and the 95% confidence range is between 3x and x/3, where x is the unit cost value. Subsequently we applied three unit levee investment cost prices (low: USD 2.33 million, mid: USD 7.0 million, high: USD 21 million) in line with previous studies[42,90]. These cost estimates were then adjusted for all other countries by applying construction index multipliers (based on civil engineering construction costs[91]), to account for differences in construction costs across countries[92]. Costs were converted to USD$_{2005}$ power purchasing parity (PPP), to be consistent with the SSPs, using GDP deflators from the World Bank (https://data.worldbank.org/), and annual average market exchange rates between Euros and USD taken from the European Central Bank (unit levee cost per country = unit levee cost x construction index per country / PPP MER rate 2005 index per country). Example: mid unit levee costs$_{USA}$ = 7.0 ×1 / 1 = 7.0 million USD$_{2005}$ PPP km m$^{-1}$. If for a country data was not available in the database, we used the average of all countries in the same World Bank income group. For the reference year 2005, this applies to Western Sahara (ESH), North-Korea (PKR) and Somalia (SOM).

**Reliability**. A scoring table was used to get insight in the reliability of the results of the global analysis. Results were grouped into four reliability classes ranging from "poor" to "very good". Transects were placed in these classes based on data accuracy for three characteristics: hydrodynamics, vegetation and profile elevation. In Supplementary Fig. 6 the (sub) results of the analysis are presented. The first category, hydrodynamics, included known inaccuracies in the hydrodynamic data (GTSM and ERA-I). Data from the GTSM model was considered less reliable in areas with a low tidal range and/or with tropical storms, such as cyclones or hurricanes, as those were not included in our analyses. Also wave data from ERA-I are less reliable in these areas, because the effects of tropical storms are flattened due to the relatively coarse grid size. Hence, transects in these areas were pinpointed by linking them to NOAA data of historical hurricane tracks[93]. In Supplementary Fig. 6B, areas where tropical storms occur can clearly be recognized. In addition, the Mediterranean Sea, the Red Sea, the Black sea and the Caspian sea stand out in inaccuracy, because of limited tidal action.

Reliability of vegetation characteristics was determined by data source and vegetation width. For transects with extensive vegetation widths, crest height reduction was less sensitive for possible deviations of the vegetation width, due the non-linear relation between vegetation width and wave reduction. Vegetation cover proved most reliable in areas where data from the salt marsh[32]—and mangrove map[14] were available. Hence, this resulted in a 'good' score (Supplementary Fig. 6C). Only in cases of extensive vegetation presence was a 'very good' score assigned. Transects were appointed as "very good" if vegetation extended 500 m for mangroves, and 1000 m for salt marshes. These thresholds are chosen based on our model results, which show that after ~500 m (salt marshes) and 1000 m (mangroves) maximum reduced wave transmission by foreshore vegetation is reached. Vegetation cover reliability in Europe was classified as 'good', due to reliable vegetation type classification based on CLC[30] and the salt marsh map[32] in combination accompanied by relatively small vegetation widths. The reliability of the derived vegetation characteristics is especially lacking at the east coast of Canada, at Latin America's south coast, at Africa's coasts facing the Mediterranean Sea, coasts along the Red Sea and the Persian Gulf, and along the coasts of China, Japan and Russia. For example, in the Persian Gulf states the vegetation presence map tends to falsely identify foreshores as vegetated.

The time-ensemble average (TEA) technique applied for the FAST intertidal elevation map relies on the availability of a reasonable number of images at different tidal stages where the differences in horizontal extent of water coverage can be identified, thus allowing a composite of inundation frequency to be derived. However, the technique is limited by the effective sensor resolution (~30 m, including uncertainty in georeferencing) relative to the horizontal extent of changes in inundation, a function of the tidal range and bed slope. Hence, changes in tidal water extent in microtidal or very high bed-slope regions tend to be too small for reliable discerning differences, leading to poor performance of the technique. However, the merged GEBCO-MERIT dataset was considered less reliable than the FAST intertidal map, based on the resolution and the merging of the two underlying datasets in the intertidal zone. In addition, MERIT tends to overestimate the elevation in mangrove areas, as it measures the canopies as the earth's surface. Besides the elevation data, the foreshore definition method is used as a profile reliability indicator. The total score per transect is given by the sum of the sub-scores. The sub-scores are normalized to give equal weight to the scoring categories.

**Validation.** For validation of our method to assess vegetation presence, a comparison of 280 randomly located transects with aerial imagery was carried out. The area accessed in the global assessment was divided in tiles of 90 degrees longitude and 15 degrees latitude. From each tile 6 vegetated and 2 non-vegetated transects were selected. Next, a reference dataset was created by manually identifying vegetation presence using present imagery. Lastly, the vegetation width derived by the model and the manually derived set were compared (Supplementary Fig. 8). For this comparison we made three distinctions, based on (1) vegetation type, (2) foreshore derivation method and (3) vegetation cover source. Comparison showed that the used algorithm on global EO data performs satisfactorily (Supplementary Fig. 8), but in some cases tends to assign a vegetation cover of up to 250 m where there is none. Deviation between observation and the global assessment, is caused by methodological error in the global assessment and inaccuracy in the global datasets, e.g. different timestamps are inevitably compared. This would induce an exaggeration of the effect of vegetation. However, due to the limited dimension of the vegetation extent, the threshold for substantial crest height reduction is falsely exceeded in not more than 2.4% of the cases and the effect is largely balanced out by underestimation of the vegetation cover at larger lengths.

To validate wave reduction by vegetation calculated through our lookup table approach, we compared results with local modelling results for the South-Western part of the Netherlands for 38 vegetated transects. The numerical model SWAN[94] in stationary mode was used to translate wave conditions from offshore to nearshore. The simulations were performed with a grid size of 0.01 deg and bathymetry from EMODNET[95]. Extreme water levels were included by a water depth correction, using data from GTSR[18]. Both wind and wave boundary conditions were derived from the earlier described ERA-I re-analysis. The governing wave direction was based on the average of the fifteenth highest wave events in the available wave data. The wind direction was assumed to be aligned with the wave direction. A parametric JONSWAP spectrum shape was used, using a peak enhancement factor of 3.3 and directional spreading of 20 degrees. Foreshore profiles were constructed using an approach similar to foreshore method 2 in the global study but using local high-resolution bathymetry and topography data. Vegetation width was extracted from the salt marsh map[32], which was confirmed locally using aerial imagery. Foreshore wave propagation was determined using XBeach in surfbeat mode[79].

Our results showed an overestimation of the water depth at the start of the vegetated zone by 0.73 m on average. In addition, the global model derived milder slopes in comparison to the local analysis for narrow vegetated transects. The largest errors were found further away from the mouth of the estuary. Here, the deviation between the wave calculated by SWAN and the depth limited approach is largest. The wave height at the start of the vegetated zone was overestimated on average by 1.12 m, due to the complex geometry and the sheltered configuration of the estuary. The algorithm approximated the wave transmission reduction (RMSE

13%) and the levee crest height reduction relative to the required crest height without vegetation presence (RMSE 19%) with reasonable accuracy (Supplementary Fig. 9).

**Sensitivity analysis.** A sensitivity analysis has been performed to provide insight in the uncertainty in the presented potential global levee costs savings. The analysis focused specifically on single key parameters, such as the levee unit cost, the critical overtopping discharge and the wave breaker index. High, mid and low levee unit cost scenarios are taken from previous studies[42,90]. A high, mid, low for the critical overtopping discharge are respectively 10, 1 and $0.1 \, \mathrm{l \, s^{-1} \, m^{-1}}$ to incorporate the quality of the levee cover[47]. We chose RP10 and RP1000 for, respectively, the low and high storm return period scenario. The uncertainty spread of vegetation width is based on the 75% confidence intervals of the underestimated and overestimated vegetation widths of mangroves (+436 m, −136 m) and salt marshes (+597 m, −104 m) in the vegetation presence validation study. For the breaker index we solely chose a high scenario of 0.78, because the index of the global assessment (0.55) was already quite conservative[71]. For topography we applied a range corresponding to the typical vertical accuracy of the FAST intertidal elevation dataset (±50 cm). Two representative subsets of 500 transects for respectively mangroves and salt marshes have been derived using the clustering method k-means[96], based on hydrodynamic conditions, vegetation cover, profile characteristics and geographical location. With these subsets, we repeated the analysis procedure of the global assessment for the sensitivity scenarios. The results point out that the largest spread is caused by the uncertainty in the unit levee cost with −66% and +200% for, respectively, the low and high scenario with respect to the global reference analysis. The other scenarios: topography (−39%, +47%), critical overtopping discharge (−40%, + 40%), storm return period (−28%, +34%), vegetation width (−28%, +39%), breaker index (+21%) (Supplementary Fig. 10). Larger water depths result in a decrease of depth-induced wave energy dissipation and more dissipation due to wave-vegetation interaction, which explains the outcomes of the topography sensitivity results. Similarly, an increase of the storm return period or the breaker index shifts the ratio of wave energy dissipation by wave-bottom interaction and wave-vegetation interaction. The coastal protection costs by vegetation are sensitive to critical overtopping discharge changes, because of the non-linear relation between the wave height in front of the levee and the overtopping discharge[47].

## Data availability
Data generated in this study (Data for Figs. 1, 2 and 3) have been deposited in the Zenodo database under accession code https://doi.org/10.5281/zenodo.5120632. The FAST intertidal elevation map and the FAST vegetation presence map data are available under restricted access due to the large size of the dataset, access can be obtained freely upon reasonable request to the corresponding author.

## Code availability
All general-purpose software packages that we used are open source: Python 2.7.14 (https://www.python.org), NumPy (http://www.numpy.org/), GeoPandas (http://geopandas.org/), Xarray (http://xarray.pydata.org/en/stable/), SciPy (https://www.scipy.org/), Rasterio (https://rasterio.readthedocs.io/en/latest/), Shapely (https://pypi.org/project/Shapely/). Maps were created using Cartopy (v0.18.0. Met Office UK. https://pypi.python.org/pypi/Cartopy/0.18.0) and Matplotlib[97] v3.3.4. The software written specifically for this project is available from the corresponding author on reasonable request. The numerical models used in this study are available at: (XBeach) https://oss.deltares.nl/web/xbeach/source-code-and-exe, (SWAN) http://swanmodel.sourceforge.net/.

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

## Acknowledgements

The research leading to these results received funding from the Aqueduct Global Flood Analyzer project, via subsidy 5000002722 from the Netherlands Ministry of Infrastructure and the Environment. The latter project is convened by the World Resources Institute. This research is part of Foreshore Assessment using Space Technology (FAST, 2014–2018). A project funded by the European Union's (EU) Seventh Framework Programme (FP7) for research, technological development and demonstration under grant agreement number 607131. FAST is developing downstream services for the European Earth Observation Programme Copernicus to support cost-effective, nature-based shoreline protection against flooding and erosion. P.J.W. received additional funding from the Dutch Research Council (NWO), in the form of a VIDI research grant 016.161.324.

## Author contributions

V.T.M.Z. wrote a substantial part of the manuscript, wrote and ran the (post) processing pipeline and produced all figures unless otherwise indicated. J.T.D. participated in conceiving the conceptual method, wrote a substantial part of the manuscript, setup the preliminary version of the final processing pipeline, wrote parts of the final processing pipeline. B.K.W. participated in conceiving the conceptual method, wrote a substantial part of the manuscript and provided photos for Fig. 4. D.E. wrote (major) parts of the processing pipeline and contributed to the manuscript. E.P.M. developed the FAST intertidal elevation map, assisted with the FAST coastal vegetation map, contributed to the manuscript and produced Supplementary Fig. 7. H.C.W. Conceived experiments and contributed to the manuscript. P.J.W. provided construction index multipliers and GDP deflators at country level for levee costs calculations and contributed to the manuscript. M.B.V. participated in conceiving the conceptual method and contributed to the manuscript.

## Competing interests

The authors declare no competing interests.
