## [Peer Review File · Nature Communications]

Cutting the costs of coastal protection by integrating vegetation in flood defences.Reviewers' Comments:

Reviewer #1:

Remarks to the Author:

Dear authors,

I reviewed your interesting manuscript. It introduces a valuable dataset based on remote sensing providing intertidal elevation and a coastal vegetation map. However, the transposition of those datasets and results of hydrodynamic modelling into coastal protection requirements has many issues, which I will outline below. The study relies on a long chain of models, datasets and assumptions, therefore I will focus on the Methods section, as the main text is generally well-written.

The introduction makes the impression that only the high-resolution intertidal elevation is used (l. 43), but the methods (l. 101) state that it was used only in 47.6% of coastline. Also, the authors write that the dataset has "typically a 30-50 cm vertical accuracy", but in Extended Data Fig. 7 both RMSE and MAE are outside this range in all cases (except RMSE in panel 3C). The error is also mostly proportional to the variation of height. Then, the GEBCO dataset is described as having a vertical accuracy of "tens of metres" (l. 55) and MERIT of 2 m. Given that the GEBCO/MERIT is actually used in majority of profiles, the average levee crest reduction was 30 cm, and the wide differences between the two elevation datasets shown in Extended Data Fig. 2, a validation of GEBCO/MERIT would be very much needed. I'm not sure how much the profile's slope matters here on the results, but the validation study (Extended Data Fig. 9) shows significant overestimation of wave heights and resulting crest height reduction. A broader sensitivity analysis could reveal the degree of uncertainty in the results.

The description of how vegetation is brief and not too informative. The authors write that "[d]etermining global coastal vegetation extent is difficult and affected by various idiosyncrasies, including algae presence" (l. 62-63). How did the authors overcome those problems and how much could this affect the results?

Wave heights in the nearshore are transformed by the authors using a single global assumption of the breaker criterion (l. 77). What is the source of this value and how sensitive are the results on this assumption?

In the section on vegetation parameters (l. 124-137) no literature is cited to support the choices of model parameters, even though "literature" is mentioned twice.

Parts on profile construction, wave attenuation models and levee crest heights contain necessary and reasonable simplifications, but then the assumptions on the actual coastal protections needs is far too simple and therefore greatly overestimates the reduction of protection costs. Firstly, if I read the methods correctly, no topographic features protecting the land from flooding are considered, but a need for a full levee is assumed everywhere. Then, the authors use a single protection standard (why 100 years?), but the actual needs will vary significantly e.g. between rural and urban areas, which are distinguished in the paper. Though it is shown that the results are not very sensitive to the choice of return periods (main text l. 178), this aspect combined with neglect of actual topography (even if not very accurate) results in very large global needs for protection. Finally, protection is assumed to be needed everywhere, even in places not populated or used economically. This is probably why the protection savings in Australia are so huge (almost 55 billion USD), whereas the vast majority of the coastline is uninhabited, so there is nobody to actually benefit from most of the savings. The same effect is present in other countries indicated to make large savings, especially Canada and Russia. The analysis should be limited only to areas where floods can cause actual losses to people or assets, given the terrain and exposure. This would probably revise down the savings enormously. It also would make the coastal risk analysis (Fig. 3) redundant.

The authors write (l. 174-175) that a one-meter heightening of levees was shown in several studies to cost 7 mln USD. Two of the referenced publications were not accessible to me, but the other publications show that levees cost 8-9 mln USD (Aerts et al. 2013), 0.6-22.4 mln USD (Jonkman et al. 2013) and 0.01-25.4 mln USD (Lenk et al. 2017). The uncertainty of the costs of the levees translate very directly into the paper's headline result of 194 billion USD cost reductions. Further, the authors seem to first convert US and Dutch levee costs first by international differences in construction costs (referenced publication was not accessible) and then further with GDP deflators (l.

179–181). Why this double conversion done and why with 2005 PPPs when newer 2011 PPPs have been long available? Then, the authors show the total saving which refer to unknown year (Fig. 2A) and then relative to GDP from unknown year (Fig. 2B). Also, savings are shown for countries for which PPPs are not available, such as Somalia and North Korea. How it is possible given the methodology? In summation, the authors should account for uncertainty in levee construction costs, and precisely define the economic data and their sources used to arrive their headline results.

The analysis of the reliability doesn't add much to the confidence into the results, as it is difficult to connect it with country-level information. At the very least, this should be somehow indicated in Fig. 2, but a broader incorporation of uncertainty (construction costs, topography, wave breaking parameter etc.) would be better.

Regarding the discussion, I also wonder how much the benefit from levees behind marshes or mangroves would be reduced by the problem of subsidence in such unstable grounds. In Poland, in the marshy coastal areas subsidence of levees is a problem, resulting in extra height added to levees to maintain the required safety standard over time. Engineering levee designs included the possibility of 20–70 cm subsidence in such an area which I researched myself.

The study also largely ignores the problem of sea level rise. Wouldn't SLR permanently destroy the low-lying vegetation in many places, reducing wave attenuation and therefore the present cost savings would be lost in the near future?

Minor points:

The study area is defined either as between "the polar circle and -60° S" (main text l. 62-63) or "the Arctic and Antarctic circles" (methods l. 31). Polar circles are around both poles, and they are not at 60° latitude. Precise latitudes should be added in both places.

The maps in Fig. 1 make an impression that most of the coast benefits from vegetation, which is not true. Non-benefiting coasts should be also marked, unless it obscures the map too much. The color scale in 1A should be changed to follow a more logical progression like in Fig. 2. In Fig. 2B the Caspian Sea is shown as "no data". Also, the land-locked countries should be highlighted with grey color, not those that benefit only insignificantly.

In conclusion, the paper is strong on the remote sensing and hydrodynamic aspects, and those aspects are a very interesting development. However, the coastal protection part has many issues that need to be addressed in order to provide a robust result.

Kind regards,
Dominik Paprotny

Reviewer #2:

Remarks to the Author:

In their paper entitled "Cutting the costs of coastal protection: how vegetation reduces global flood hazard," van Zeist and colleagues model the coastal hazard mitigation provided by mangroves and marshes on a global scale. They find that nearly a third of the global coastline is covered with vegetation and that this vegetation considerably lowers required levee height resulting in large cost savings. The analysis is based on global process-based models of wave evolution and new higher resolutions intertidal elevation datasets that make this wave modeling possible. The paper is an important contribution to the field -- specialists will be interested in reading it -- but I do have some reservations about publication in Nature Climate Change in its current state.

1) The abstract states that the contribution of ecosystems to coastal flood risk has never been quantified at a global scale. This is misleading and reflects that the authors flood risk modeling perspective. They may be less familiar with the ecosystem services literature.

Beck et al 2018 Nature Communications quantify flood risk reduction provided by coral reefs globally using a similar modeling approach to this paper. Chaplin-Kramer et al. 2019 Science use a coastal hazard index to quantify the coastal risk reduction provided by corals, marshes, mangroves, and other coastal forests, as well as two other ecosystem services. Several studies including and following on Costanza et al. 1997 use a benefit transfer approach to quantify the value of coastal protection

services at a global scale. And Hochard et al 2019 use an empirical, statistical approach.

What is novel about this work is that it employs a process-based model for quantifying wave attenuation provided by vegetation, leveraging the newly available elevation data and accounting for spatial variation in key biophysical and economic variables. So it's more mechanistic than previous studies, which is important for understanding factors influencing mitigation and where to direct investments in restoration and conservation. I suggest the authors put the advancements of their study in the context of the previous ES literature as well as any relevant flood risk reduction papers.

Another minor point is that the ecosystem services literature has few examples of urban ecosystem service analyses of coastal protection. This analysis explicitly breaks down urban and rural areas so it could in theory assess mitigation provided by vegetation for urban areas and elevate that point.

2) One issue with their approach to quantifying the reduction in costs is that it assumes levees at the end of the vegetation. It may be worth discussing limitations to this climate adaptation strategy. Would levees prevent the mangroves from migrating as sea-levels rise? What about sources of sediment to allow for accumulation?

3) Another issue that merits a least a couple of sentences is the replacement cost approach the authors use to quantify mitigation benefits. See Barbier et al. 2015 Ecosystem Services for a discussion of the limitations of the replacement cost approach. Economists generally think the avoided damages approach is superior. The RC approach estimates a benefit as a cost, human built alternatives are rarely more cost effective, and the RC approach makes the assumption that the alternative WOULD be built when that might not in fact be the case, especially at the global level and given the costs in remote places. So there are drawbacks of the economic approach used in this paper that should be discussed.

4) The paper uses the well-regarded model, XBeach for the wave attenuation analysis. But it doesn't report the equation in the model accounting for attenuation in vegetation. That should be in the methods since it's a key aspect of the mechanistic approach. Also the analysis uses a single value for key parameters like height of mangroves etc, thus it doesn't capture variation in the attributes of vegetation even though it is capturing spatial coverage. This variation could have importance consequences for cost reduction provided by vegetation and is a limitation that should be mentioned.

5) Legend in Fig 3 is confusing. What is meant by "high potential countries"? I follow the logic of including the percentage of population exposed in the risk reduction score per country, but I would also point out that the authors are mixing a risk approach (exposure/consequence) with a quantification of costs/levee height reduction approach. Did the authors use any theory / conceptual model to base this on? The conceptual model and analysis for the biophysical part of the modeling is strong, but the links between the biophysical and economic/social side and the conceptual model for economic analysis are weaker.

6) In some places there seems to be more jargon (e.g., "required crest heights for 100-yr protection standard" than appropriate for an interdisciplinary journal such as Nature and the results are very detailed with a lot of % rather than compelling descriptions of the magnitude of differences in words. This struck me as also less appropriate for a high impact journal such as Nature CC.

In addition, I found some colloquial language such as "profits will for sure result" on line 236, and lots of missing words or incoherent lines that suggest the paper needs a good proof read—even in the last sentence of the abstract. The authors start to make some important points in the paragraphs starting on lines 195 and 206, but these paragraphs are unclear and sentences don't necessarily segway from each other. For instance they say these ecosystems also provide other services – why is that important to this study? Grey infrastructure cannot provide ES – would info on those services inform the calculus further?

7) The authors may be interested in Silver et al. 2019 *Frontiers in Marine Science* as an example of a recent coastal protection analysis in a SIDs with long rural coastlines not likely to be protected with infrastructure (see first paragraph of the discussion).

Reviewer #3:

Remarks to the Author:

Key points:

The paper addresses an important topic in light of recent, growing attention given to 'nature-based' solutions towards coastal flood protection and mounting evidence that vegetated foreshores are of critical importance in buffering against the impact of extreme storm/wave events. However, two key points have led me to recommend rejection of this paper:

1) Methodological issues: the expectation of the authors to be able to estimate this very locally determined function of vegetated coastal ecosystems is fundamentally flawed. Waves reaching the upper intertidal zone are the result of morphodynamic interactions that begin a considerable distance seaward/below the limit of the vegetation present in the intertidal zone, particularly on shallow sloping coasts, where waves begin to be dissipated several kilometers before they reach the upper intertidal zone, such that plants can establish there. Without confidence in accurately predicting not just the intertidal topography but also the sub-tidal, nearshore topography, the application of the methods reported here is, to my mind, fundamentally problematic (at least there is no concrete evidence presented in the paper to convince the reader that the results are robust and errors are not reported for the global analysis). There are several comments below that relate to the lack of a robust methodological accuracy assessment and/or transparency of its results. Figure 6 to 8 are critical here in that they illustrate the wide spread of the data around the best-fit line between modelled and validation data. With respect to vegetation width, e.g., it is not uncommon for widths to be over or under-estimated by 50-100% which is a significant error term. The authors only explicitly state what the overall errors were (absolute and relative) in wave dissipation estimates for a particular location (Lines 244-251) but then report global estimates. They do not state what the expected accuracy is in the height estimates at the seaward margin of the vegetation globally, nor how coastal bathymetry in the sub-tidal zone is taken into account (where waves are already affected by the bed). Without this, the dissipation percentages reported here at the global scale rather lose their value (nearshore bathymetry arguably changes over similar time-scales, if not more rapidly, than the upper intertidal shore (whether through human intervention or 'naturally')). I rather suspect that errors in bathymetry and closer to shore away from the test sites can be many times larger than at the 'test-location' reported here on many transects globally, due to the sensitivity of wave conditions to nearshore (as well as intertidal) elevations. Where waves are likely to be depth limited, even the reported accuracy of 50cm in the intertidal topography is likely to be significant in terms of its impact on wave height and wave dissipation estimates.

2) Conceptual issues: The use of the term 'ecosystem' in the context of wave dissipation can be a bit misleading – it is really the landforms that are associated with the ecosystems here that fulfil much of the protecting function as they constitute an altered bathymetry in the nearshore. The results are presented with very little detailed information on context. See, for example, my comment on line 83. Presumably, the authors have determined these percentages of wetland fringed coasts and flood susceptible coasts on the basis of the 1km spaced normal coastal transects? Flood risk, particularly due to wave action, can be much more locally variable than this (as it may depend on abrupt changes to coastal landform configuration, e.g. breaching etc..). The authors should at least acknowledge this, if not comment on the implication of ignoring this. Some of the results statements appear entirely unsupported, e.g. lines 84-85: 'These vegetated foreshores on susceptible coastlines are encountered to a similar extent at both urban and rural

coastlines' – no evidence is presented to back up this statement. What criteria was applied to classify a coast as 'urban' versus 'rural'?

The discussion section makes many claims (e.g. about the benefit of coastal vegetation in the first sentence and the claim that small island development countries largely depend on wave reducing abilities of coastal vegetation) without referring to relevant literature (there is an almost complete lack of citation) or results provided in the paper itself.

Line-by-Line Comments:

Line 12: 'growing population and economy' sounds a bit strange... The economy is not necessarily growing everywhere... Do the authors really mean growing 'economic activity'?

Line 20: '... financially attractive designs.' – of what? It would help if the authors were a bit more explicit here.

Line 20-22: 'We calculated that...' and also Line 75: 'For areas with no levees...' – to present this suggestion rather uncritically here can be misunderstood. Are the authors really contemplating the establishment of levees along all coastlines susceptible to flooding? That is clearly a very short-sighted idea as there are many coastlines susceptible to flooding where it would be entirely foolish to build any kind of levee – as it is well known that the construction of levees encourages settlement and development landward of the levees and in fact may increase the population and asset value at risk from flooding, even though it reduces the likelihood of flooding over the short term. Even intertidal ecosystems will eventually cease to protect the levee, where sea level rise and lack of sediment delivery constrain the future existence of the ecosystem....

Line 30: 'purely grey solution' – the 'grey' should be in inverted commas.

Line 31: refers to 'ecosystem services' but the text before this only identifies ONE ecosystem service. The authors should elaborate on the many other ecosystem services provided by coastal ecosystems.

Line 31: the sentence starting 'Despite...' sounds as though providing ES should be protecting wetland areas from decreasing... Please rewrite.

Line 32: I think it should be 'Over the short term...' and 'over the long-term...' and the authors should give an indication of what they mean by either. The point about 'loss of coastal lands' is a little naïve – as the loss of coastal land is part of naturally dynamic coasts that erode in some places and form new landforms in others. There is no recognition in here of this.

Line 49: The authors state that '...the effect of coastal wetlands on storm surge levels is typically only relevant for very large wetlands and still leads to flooding in areas that are not protected by levees'. This statement seems confusing without any further explanation here. Are the authors saying that coastal wetlands can exacerbate flooding in areas that are not protected? Or are they simply saying that coastal wetlands reduce storm surge levels but not where there are no wetlands (which seem rather obvious...)? This needs rewriting/clarifying.

Line 52: What is meant by 'adaptive designs' here? Adaptive to what precisely? There are many changes that one might want to adapt coastal protection to...

Line 54/55: This is a good example of where the authors are using a rather unclear and unhelpful written style: '... at the end of the vegetated foreshore' – where would the 'end' of a 'vegetated foreshore' be? Is it the Highest Astronomical Tide level? With or without extreme meteorological forcing?

Line 59: What are the 'existing vegetation maps' referred to here?

Line 62/63: surely the 'polar circle' is either North or South pole, so the 'and -60 S' is not clear here.

Line 61: The ERA-Interim data ocean wave model horizontal resolution is 110 km with wave spectra discretised using 24 directions and 30 frequencies – with many coastal wetland coasts extending in length for less than this distance and often situated in topographically complex embayments or estuaries, it is not clear here how the authors (a) interpolated to finer resolution to capture the effect of such wetlands on such coasts and (b) took account of wave refraction/reorientation over complex bathymetry – something that may lead to wave dissipation (or at least along-shore redistribution of wave energy) from the offshore to the nearshore regions in its own right. It is also unclear whether the models accounted for wind direction at the time of wave modelling. In the shallow environments in

which coastal wetlands grow, waves are likely to be depth limited by the time they reach the wetland. They lose a significant amount of energy due to breaking in the process and whether winds are blowing on- or off-shore can make a significant difference to wave height. The authors do not comment on this.

Line 67/68: In considering this comparison between the foreshore with and without vegetation, the authors need to consider that the landform that determines the bathymetry and on which the vegetation is growing is itself the result of the presence of the vegetation – i.e. vegetated coastal wetlands are formed by both externally derived (allochthonous) and internally derived (autochthonous) organic material through plant growth.

Line 83: ‘...27% is susceptible to coastal flooding’ – the authors do not say over what time period? How was this figure arrived at?

On line 95, the authors refer to the ‘non-linear relation between wave attenuation and vegetation width’. In fact, the literature has shown that significant wave attenuation occurs over only a 40m distance of vegetated coastal wetlands and certainly over an 80m distance, after which little wave energy is left. The authors state that 13.1% of coastal wetlands occupy a cross-shore width of 0-100m (figure 1A) but it seems that this class includes ‘0’, i.e. it is not clear what percentage has a width that exceeds a width over which any degree of wave dissipation can be expected. How accurate the width estimates? What error margin is involved in these estimates?

Line 159: ‘... where vegetation protects most assets and people’ – vegetation alone does not protect assets and people, thus this needs to be changed to ‘vegetation helps to protect....’...

Line 161: sentence does not make grammatical sense.

Line 167-168: this mentions errors but does not give error bands or any concrete information on them. ‘...some results may be under- or overestimated’ is simply far too vague for a scientifically credible paper and journal.

Line 171: what is meant by ‘slightly underestimated’? See comment above – this needs to state the quantitative uncertainty estimates here to be credible. The same applies to any other reference to errors here (e.g. ‘an underestimation of wave attenuation’ – by how much? – then (on line 174) ‘an overestimation of wave attenuation results’ – by how much?). How credible are the full study results if there are significant under and over estimations reported?

METHODS:

Line 71-74: the authors state “SLR and subsidence were not taken into account because this study focuses on the present situation. Moreover, quantifying the future role of vegetated foreshores would not only require sea level rise scenarios but also an insight in the development of wetlands over time, which is strongly determined by local conditions such as sediment supply.” – but (1) no references are given to back up the latter statement and (2) these constraints are not highlighted sufficiently in the main body of the paper and its concluding section (thus not providing a sufficiently balanced and cautionary approach towards the full implications of the results of this study – something that is of utmost importance to a high-impact journal such as this).

Line 77: Here and in the results and discussion, the authors never state what the distribution of above-wetland wave heights was that this modelling resulted in. The breaking criterion and the fact that vegetated wetlands (marshes and mangroves) only exist at the very landward limits of the intertidal profile, results in relatively small wave heights at the seaward margin of the vegetated zone. It also means that the unvegetated surfaces seaward of the vegetated wetland are just as important at re-distributing and dissipating wave energy as, without such re-distribution/dissipation, the wetland would most likely not (be able to) exist. The authors rather gloss over this in both discussion and methods.

Line 82 onwards: Profile construction: The key issue here relates to my comment above: the authors frequently here make statements such as ‘the accuracy [...] required...’ or ‘validity checks were performed...’, or ‘a check was performed...’ without giving detail on what the exact criteria were for discarding/accepting data and what the implications for accuracy/error were overall as a result. What was the estimation/modelling error for water depth? What was the estimation/modelling error for wave heights along the transect? It is extremely important to communicate the relative magnitude of

the estimated values alongside their errors. The authors only report on percentage wave height change but even this is not communicated clearly with error margins.

Line 128-130: The authors state "The modelling approach uses four parameters to represent vegetation: height, diameter, number of stems and drag coefficient. The exact characteristics are based on observations in literature ($N = 30 \text{ m}^{-2}$, $d = 35 \text{ mm}$, $h = 3.0 \text{ m}$)." – but again, no reference is made to where in the literature (what literature) these figures come from. Surely, these parameters vary very widely around the world for different wetland types at different stages of evolution/growth?!?

Line 134: "A dragcoefficient (CD) of 0.19 is chosen, which is the lower limit found during large scale flume tests." – another statement with no reference to the literature and/or indication of how varied this parameter can be expected to be across the wetlands included in this study.

Line 160 onwards: the authors do not define what they mean by 'susceptibility to flooding' accurately enough. Is this just a definition by elevation of the coastal hinterland? Surely, it matters what the configuration of the coast is as a three-dimensional space and, as the authors themselves acknowledge earlier on, what is currently present between the intertidal vegetation and the landward areas that are 'susceptible to flooding'.... More information is needed here and a more critical / objective / comprehensive definition.

Line 184 onwards: Reliability: the authors introduce the 'scoring classes' but do not define what the actual or RMS errors are that equate to these classes.

Line 209: This is the first time 'FAST TEA' is mentioned. What does it stand for?

The figures are generally helpful, although as stated above, it would have been useful to have the actual wave heights at the seaward margin of the vegetation presented in a figure to show how high incident waves were on these types of coasts prior to being dissipated - i.e. what magnitude of a problem is the paper addressing?

Point-by-point reply

revision Cutting the costs of coastal protection

Date

16 April 2021

Contact person

Vincent van Zelst

Number of pages

1 of 42

E-mail

Vincent.vanZelst@deltares.nl

Reviewer #1

General comments:

Dear authors,

I reviewed your interesting manuscript. It introduces a valuable dataset based on remote sensing providing intertidal elevation and a coastal vegetation map. However, the transposition of those datasets and results of hydrodynamic modelling into coastal protection requirements has many issues, which I will outline below. The study relies on a long chain of models, datasets and assumptions, therefore I will focus on the Methods section, as the main text is generally well-written.

Reply:

Reviewer 1 (Dominik Paprotny) expressed his interest in the manuscript. The reviewer specifically highlighted the strong remote sensing and modelling part, however the reviewer has concerns on the coastal protection part, especially the assumptions on coastal protection needs. The comments are focused on the methods section. We like to thank the reviewer for his kind words and his constructive feedback. Based on the review we:

- Reconsidered our assumptions on coastal protection needs
- Extended the analysis by taking into account population density
- Included a sensitivity analysis
- Provided clarification in the main text and methods where indicated necessary by the reviewer
- Updated existing and added new figures

Hereafter a copy of the review by Reviewer #1 and a point-to-point reply on the raised issues.

Bathymetry and elevation

Question 1:

The introduction makes the impression that only the high-resolution intertidal elevation is used (l. 43), but the methods (l. 101) state that it was used only in 47.6% of coastline.

Reply:

We had to make balance between the amount of detail in the main manuscript and the methods section. Nevertheless, we do not want to provide a false sense of accuracy either. We added a sentence in the revised manuscript describing that the high-resolution datasets are combined with open-source global data layers (l. 59). In what cases the high-resolution data and the open-source bathymetry and elevation is used was already described in the methods (Methods l. 68). In addition, we updated the stated percentages in the Methods based on the filters (Methods l. 147) we applied on the global results to make the results more socio-economic relevant (point raised by multiple reviewers).

Question 2:

Also, the authors write that the dataset has “typically a 30-50 cm vertical accuracy”, but in Supplementary Fig. 7 both RMSE and MAE are outside this range in all cases (except RMSE in panel 3C). The error is also mostly proportional to the variation of height. Then, the GEBCO dataset is described as having a vertical accuracy of “tens of metres” (l. 55) and MERIT of 2 m. Given that the GEBCO/MERIT is actually used in majority of profiles, the average levee crest reduction was 30 cm, and the wide differences between the two elevation datasets shown in Supplementary Fig. 2, a validation of GEBCO/MERIT would be very much needed.

Reply:

Supplementary Fig. 7 contains three panels. Panel 1 and 2 contain a comparison of the newly derived intertidal elevation set with DTMs for two location in the USA. The errors for these panels are respectively: RMSE1 = 0.14 m, MAE1 = 0.11 m, RMSE2 = 0.24 m, MAE2 = 0.16 m. Panel 3 contains results for validation sites in the UK (A- RMSE = 1.15, MAE = 0.97), The Netherlands (B- RMSE = 0.7 m, MAE = 0.56 m) and Spain (C – RMSE = 0.36 m, MAE = 0.29 m). The values for the sites in the United Kingdom and the Netherlands are indeed outside the stated vertical accuracy range, however the average RMSE and MAE are 0.518 m and 0.418 m respectively.

The line in the main manuscript stating the typical accuracy is replaced by the actual RMSE based on the validation study sites (l. 58). To put in context, local derived DTMs derived with Lidar have often a stated vertical accuracy of +/- 0.2 m and global elevation (MERIT, 2 metres vertically) and bathymetry (GEBCO), tens of metres vertically). This indicates that using the intertidal dataset for about 50% of the transects is an enormous improvement in comparison to

global bathymetry (e.g. GEBCO) and elevation (e.g. MERIT and SRTM) datasets that are commonly used in global studies such as the *Nature* publications by (Menéndez et al. 2020; Muis et al. 2016; Vousdoukas et al. 2018, 2020). The reviewers request to validate GEBCO/MERIT seems to be not applicable as these datasets are already validated (Becker et al. 2009; Yamazaki et al. 2017) and are the most commonly used topographic datasets in global studies. In addition, the merged data layer GEBCO-MERIT applied in this study is also used in (Athanasiou et al. 2019, 2020), where this merged set is validated. Nevertheless, we agree with the reviewer that accurate bathymetry and elevation data is needed to improve predictions further. This is the main driver why we initiated the development of the global intertidal dataset. For now, we used the best available global data layers.

Question 3:

I'm not sure how much the profile's slope matters here on the results, but the validation study (Supplementary Fig. 9) shows significant overestimation of wave heights and resulting crest height reduction. A broader sensitivity analysis could reveal the degree of uncertainty in the results.

Reply:

The profile's slope influences the balance between, and the magnitude of, wave energy dissipation processes at the foreshore. To get insight in the sensitivity of the results to the topography an additional sensitivity analysis is performed. The approach of the additional analysis is described in the Methods (l. 260-282) and the results are summarized in the additional Supplementary Fig. 10. In short, larger water depths result in a decrease of depth-induced wave energy dissipation and an increase in dissipation due to wave-vegetation interaction.

Vegetation parameters

Question 4:

The description of how vegetation is brief and not too informative. The authors write that "determining global coastal vegetation extent is difficult and affected by various idiosyncrasies, including algae presence" (l. 62-63). How did the authors overcome those problems and how much could this affect the results?

Reply:

Focusing on limiting the length of the Methods section we inevitably had to keep certain issues unaddressed. We like to split the reviewer's comment on this matter in two parts: (1) choice of vegetation properties for process-based modelling, (2) determination of vegetation presence. We clarified both points in the manuscript (Methods l. 115-132 (1) and Methods l. 35-49 (2)).

1. Correct vegetation characterization is important for accurate modelling of wave-vegetation interaction. This global study assessed wave attenuation by mangroves and salt marshes. Below in short the sources from which vegetation characterization is derived.

Salt marshes

Salt marsh characteristics are derived based on (Jadhav, Chen, and Smith 2013; Möller 2006; Möller et al. 2014; Vuik et al. 2016) and field observations at the FAST field test sites in Romania, UK, Spain and the Netherlands (<https://zenodo.org/record/581247#.XtkSljzbs0>). The latter provided data on seasonal variation of the leaf area index of salt marshes in temperate areas. The data showed a clear peak of the LAI in summer. We selected a parameterization representative for a winter state as found in NW Europe, which is a conservative assumption. The applied drag coefficient (required for numerical modelling of wave-vegetation interaction) was set to 0.19 based on large scale flume tests (Möller et al. 2014).

Mangroves

Mangrove characteristics are derived based on field observations by (Brinkman 2006; Cole, Ewel, and Devoe 1999; Horstman et al. 2014; Krauss, Allen, and Cahoon 2003; Mazda et al. 1997; Narayan et al. 2011; Zhang, Chua, and Cheong 2015). From these sources we derived a characterization of (fringing) pioneering mangroves.

2. To determine vegetation presence, we developed the Foreshore Assessment using Space Technology (FAST) coastal vegetation map using Landsat-8 and Sentinel-2 satellite imagery for the period 2013-2017. Vegetation presence was obtained by applying an individual NDVI threshold per tile, with a total of 25,000 tiles, based on the yearly NDVI average and NDVI amplitude. The mentioned idiosyncrasies, apply more specifically to eutrophication in coastal environments. This might bias the estimate of wetland area to being larger than it actually is. This is a known issue in the remote sensing field. The FAST coastal vegetation map is validated on a case study in Zuidgors, the Netherlands. This location is situated in the Western Scheldt that, is an eutrophic macro-tidal estuary. The NDVI outcomes of the global dataset were compared with local in-situ NDVI measurements (Figure 1) showing a good fit ($R^2=0.92$).

In our global assessment we combined the FAST coastal vegetation map with other global vegetation data sources (mangroves map, salt marsh map, CorineLandCover and GlobCover). During validation of the vegetation width we did notice false positive assignment of vegetation cover along the coasts of the Red Sea and the Persian Gulf. To limit the influence of coastal eutrophication we only assign vegetation cover to

transects in these areas if vegetation cover is confirmed by both the FAST coastal vegetation map and the Mangroves map or the Salt marshes map. In addition, globally we removed transects with an assigned vegetation width below 250 m with a vegetation type assignment by GlobCover, because the validation showed false positive vegetated transects for this group. The additional sensitivity analysis also included the effect of changes in vegetation width. The results show vegetation cover changes are subordinate to other variables such as levee schematization and costs. We are confident that using this revised approach the influence of eutrophication on the global results is limited, but we like to include the mentioning of the matter for transparency.

Question 5:

In the section on vegetation parameters (l. 124-137) no literature is cited to support the choices of model parameters, even though “literature” is mentioned twice.

Reply:

We apologize for this mistake. We added the missing references.

Figure 1 NDVI validation FAST Vegetation presence map.

Wave breaker criterion

Question 6:

Wave heights in the nearshore are transformed by the authors using a single global assumption of the breaker criterion (l. 77). What is the source of this value and how sensitive are the results on this assumption?

Reply:

The reviewer refers to the breaker criterion that determines at what depth incoming waves break. For the global assessment we choose indeed a single value. The chosen value of 0.55 is at the lower end of the spectrum of observations in the field (Camenen and Larson 2007). This implies that the chosen value is a conservative assumption. Using the more commonly found value in literature of 0.78 would result in less wave energy dissipation by breaking, thus more potential to lose wave energy by wave-vegetation interaction. We are aware that with the use of the low breaker criterion we tend to underestimate results, but we prefer to be conservative. We assessed the influence of using a higher breaker criterion in the additional sensitivity analysis. The choice for a breaker index of 0.78 compared to an index of 0.55 would increase the reduction in global protection costs due to foreshore vegetation presence by 21 % (Supplementary Fig. 10, revised manuscript).

Coastal protection needs

Question 7:

Parts on profile construction, wave attenuation models and levee crest heights contain necessary and reasonable simplifications, but then the assumptions on the actual coastal protection needs is far too simple and therefore greatly overestimates the reduction of protection costs.

Reply:

Based on the reviewer's comment we reconsidered the assumption on coastal protection needs. Previously, the results were based on the assumption that there is a protection need along all coastlines susceptible to flooding. We do fully agree with the reviewer that this assumption leads to an overestimation of the total potential needs and the projected reduction in levee costs in our initial manuscript. For this reason, we extensively revised our analysis by taking into account the population density along coastlines susceptible to flooding and by giving insight in the sensitivity of the reduction of protection costs to different return periods in the main manuscript. Below we react on the specific coastal protection needs related comments of the reviewer and elaborate on the revised assumptions.

Question 7.1:

(1) Firstly, if I read the methods correctly, no topographic features protecting the land from flooding are considered, but a need for a full levee is assumed everywhere.

Reply:

Coastal protection need is assumed for areas that are susceptible to flooding only. In the determination of susceptible coastlines topographic features protecting the land from flooding,

as visible in MERIT elevation, are included. However, due to the resolution MERIT very local features, such as levees or flood walls are likely flattened out.

Question 7.2:

(2) Then, the authors use a single protection standard (why 100 years?), but the actual needs will vary significantly e.g. between rural and urban areas, which are distinguished in the paper. Though it is shown that the results are not very sensitive to the choice of return periods (main text l. 178), this aspect combined with neglect of actual topography (even if not very accurate) results in very large global needs for protection.

Reply:

The analysis is performed for nine protection standards (l. 85). In the revised manuscript we put more emphasis on the influence on the choice of protection standard on the reduction in protection standard costs, by incorporating an additional figure in the main text (Figure 3E). The effect of actual topography is included in the determination of the susceptible coastlines and the effect of bottom roughness is included, which yields a more accurate result compared to a simpler 'bathtub' estimation (Methods l. 150-152).

Question 7.3:

(3) Finally, protection is assumed to be needed everywhere, even in places not populated or used economically. This is probably why the protection savings in Australia are so huge (almost 55 billion USD), whereas the vast majority of the coastline is uninhabited, so there is nobody to actually benefit from most of the savings. The same effect is present in other countries indicated to make large savings, especially Canada and Russia. The analysis should be limited only to areas where floods can cause actual losses to people or assets, given the terrain and exposure. This would probably revise down the savings enormously. It also would make the coastal risk analysis (Fig. 3) redundant.

Reply:

In the revised manuscript levee cost reductions are based on protection needs along *populated* coasts susceptible to flooding (according to our analysis 17% of world's coastline). In addition, we created an additional figure (Figure 3., the previous Figure 3 is removed as suggested by the reviewer). In the new figure, we show the cost reductions for multiple population density bins, levee costs scenarios and protection standards. This approach resulted indeed in a reduction of the calculated savings as indicated by the reviewer, but gives a far more realistic overview. We explicitly choose to not use a threshold population density for which it becomes (more) likely that levees will be built, as we not dare to state that any less densely populated coastlines are not worth considering protecting. Whether to protect coastal areas to flooding by building levees and to against which protection standard is in the hands of decision makers.

Lastly, the role of vegetated foreshores is not limited to unprotected areas. In areas where levees are constructed, including vegetated foreshores in the design can take away planned maintenance needs as a lower required levee is sufficient to meet a chosen protection standard. The revised manuscript gives in our view a complete overview and a realistic insight in the distribution of the potential along coasts with various population densities for multiple protection standards.

Coastal protection costs

Question 8:

The authors write (l. 174-175) that a one-meter heightening of levees was shown in several studies to cost 7 mln USD. Two of the referenced publications were not accessible to me, but the other publications show that levees cost 8–9 mln USD (Aerts et al. 2013), 0.6–22.4 mln USD (Jonkman et al. 2013) and 0.01–25.4 mln USD (Lenk et al. 2017). The uncertainty of the costs of the levees translate very directly into the paper's headline result of 194 billion USD cost reductions.

Reply:

The unit cost of 7 million USD was an estimate based on the findings of the referenced studies (l. 173-175) and was applied in previous *Nature* publications e.g. (Ward et al. 2017). However, this unit cost steers indeed the paper's headline. For the revised manuscript, we recalculated the reduction in protection costs based on low (2.33 million USD), medium (7.0 million USD) and high (21 million USD) levee costs scenarios in line with the approach of (Lenk et al. 2017; Ward et al. 2017)(Methods l. 178-181). To highlight the uncertainty of the costs of the levees we explicitly show these levee costs scenarios in Figure 3E.

Question 9:

Further, the authors seem to first convert US and Dutch levee costs first by international differences in construction costs (referenced publication was not accessible) and then further with GDP deflators (l. 179–181). Why this double conversion done and why with 2005 PPPs when newer 2011 PPPs have been long available?

Reply:

The first conversion addresses the differences in construction costs (at Market Exchange Rate) between the countries. In the second conversion the MER values are converted to PPP. We used PPP2005 because all data in the SSP database are in PPP2005. In addition, we like to be consistent with other studies, e.g. (Ward et al. 2017).

Question 10:

Then, the authors show the total saving which refer to unknown year (Fig. 2A) and then relative to GDP from unknown year (Fig. 2B). Also, savings are shown for countries for which PPPs are not available, such as Somalia and North Korea. How it is possible given the methodology? In summation, the authors should account for uncertainty in levee construction costs, and precisely define the economic data and their sources used to arrive their headline results.

Reply:

All monetary values in the figures are in billion US\$2005. For countries for which GDP deflator data was not available, we used the average of all countries in the same World Bank income group. We added clarification in the revised manuscript's figures and text (Methods I. 186-188).

Question 11:

The analysis of the reliability doesn't add much to the confidence into the results, as it is difficult to connect it with country-level information. At the very least, this should be somehow indicated in Fig. 2, but a broader incorporation of uncertainty (construction costs, topography, wave breaking parameter etc.) would be better.

Reply:

We aggregated the reliability results to country level and updated Supplementary Fig. 6. Inclusion of these results in Fig. 2 resulted in poor visualization. In addition, the earlier mentioned sensitivity analysis (Methods I.260-282) includes the subjects mentioned by the reviewer.

Question 12:

Regarding the discussion, I also wonder how much the benefit from levees behind marshes or mangroves would be reduced by the problem of subsidence in such unstable grounds. In Poland, in the marshy coastal areas subsidence of levees is a problem, resulting in extra height added to levees to maintain the required safety standard over time. Engineering levee designs included the possibility of 20–70 cm subsidence in such an area which I researched myself.

Reply:

The reviewer rises the point that when designing levees on soft soils a margin is commonly included to account for subsidence induced by e.g. the weight of the levee. In a global assessment general assumption are inevitable, the assumption of the levee location is one of them. With this study we like to focus coastal protection and in particular Nature-based flood defense efforts to those areas where it matters the most. Local assessments should reveal what the most beneficial levee location is. In addition, for locations where levees are or will be

established on soft soil a reduction in the required levee crest height due to vegetation presence will limit subsidence, as a reduction in levee height means simply less weight on the subsurface. While this is true, high subsidence rates are often linked to groundwater extraction (Nicholls et al. 2021). In the revised manuscript we noted that subsidence should be considered in a local evaluation (l. 286).

Question 13:

The study also largely ignores the problem of sea level rise. Wouldn't SLR permanently destroy the low-lying vegetation in many places, reducing wave attenuation and therefore the present cost savings would be lost in the near future?

Reply:

The current study focuses on the present day situation, but effects of SLR are mentioned throughout the manuscript. The faith of wetlands under SLR is under research and depends very much on local factors, such as sediment budgets, nutrients and vegetation type. Promising studies indicate that wetlands are able catch sediments and thereby keeping up with SLR (Kirwan and Megonigal 2013; Willemsen et al. 2016; Woodroffe et al. 2016). Loss of salt marshes is not an inevitable result of SLR as long as sufficient accommodation space is available that can be achieved by careful implementation of Nature-based Solutions (Schuerch et al. 2018). Erosion of the low-lying vegetation means a reduction in vegetation width, which (depending on the total vegetation width) will not directly result in a large reduction of the wave dissipating capacity of the wetland. The additional sensitivity analysis does indicate a minor influence of changes in vegetation width in comparison to e.g. levee schematization. Nature-based Solutions are seen as the most sustainable way to counter SLR. In the revised manuscript we elaborated further on effects of SLR (l. 254-267 and l. 284-288).

Minor points:

Question 14:

The study area is defined either as between “the polar circle and -60° S” (main text l. 62-63) or “the Arctic and Antarctic circles” (methods l. 31). Polar circles are around both poles, and they are not at 60° latitude. Precise latitudes should be added in both places.

Reply:

Precise latitudes are added at the lines pointed out by the reviewer (l. 82 and Methods l. 3).

Question 15:

The maps in Fig. 1 make an impression that most of the coast benefits from vegetation, which is not true. Non-benefiting coasts should be also marked, unless it obscures the map too much.

Reply:

While plotting the data we tried plotting also the non-benefitting coasts, but this obscures the map too much. The percentages of the coast benefitting from the coast are described in the text (l. 118-124).

Question 16:

The color scale in 1A should be changed to follow a more logical progression like in Fig. 2. In Fig. 2B the Caspian Sea is shown as “no data”. Also, the land-locked countries should be highlighted with grey color, not those that benefit only insignificantly.

Reply:

The suggestions of the reviewer are implemented in the revised manuscript Fig. 1a.

Question 17:

In conclusion, the paper is strong on the remote sensing and hydrodynamic aspects, and those aspects are a very interesting development. However, the coastal protection part has many issues that need to be addressed in order to provide a robust result.

Reply:

Based on the reviewer's comments we reconsidered our assumptions on coastal protection needs to end up with a more realistic result. In addition, we added more insight in the influence of socio-economic factors (such as population density and levee construction costs) and single key parameters (such as the breaker index). We are confident that the revised manuscript is more robust and insightful.

Kind regards,
Dominik Paprotny

Yours sincerely,

Vincent T. M. van Zelst ^a, Jasper T. Dijkstra ^a, Bregje K. van Wesenbeeck ^{a, b}, Dirk Eilander ^{a, c}, Edward P. Morris ^{d, e}, Hessel C. Winsemius ^{a, b}, Philip J. Ward ^c and Mindert B. de Vries ^a

^a *Deltares, P.O. Box 177, 2600 MH Delft, The Netherlands*

^b *Delft University of Technology, Faculty of Civil Engineering and Geosciences, P.O. Box 5048, 2600 GA Delft, The Netherlands*

^c *Institute for Environmental Studies (IVM), Vrije Universiteit Amsterdam, 1081 HV Amsterdam, The Netherlands*

^d *Instituto Universitario de Investigación Marina (INMAR), University of Cádiz, 11510 Puerto Real, Cádiz, Spain*

^e *EO4 Data Science, 11500 El Puerto de Santa Maria, Cadiz, Spain*

References reply reviewer #1

- Athanasίου, P., A. van Dongeren, A. Giardino, M. Voudoukas, S. Gaytan-Aguilar, and R. Ranasinghe. 2019. "Global Distribution of Nearshore Slopes with Implications for Coastal Retreat." *Earth Syst. Sci. Data* 11(4):1515–29.
- Athanasίου, Panagiotis, Ap van Dongeren, Alessio Giardino, Michalis I. Voudoukas, Roshanka Ranasinghe, and Jaap Kwadijk. 2020. "Uncertainties in Projections of Sandy Beach Erosion Due to Sea Level Rise: An Analysis at the European Scale." *Scientific Reports* 10(1).
- Becker, J. J., D. T. Sandwell, W. H. F. Smith, J. Braud, B. Binder, J. Depner, D. Fabre, J. Factor, S. Ingalls, S. H. Kim, R. Ladner, K. Marks, S. Nelson, A. Pharaoh, R. Trimmer, J. von Rosenberg, G. Wallace, and P. Weatherall. 2009. "Global Bathymetry and Elevation Data at 30 Arc Seconds Resolution: SRTM30_PLUS." *Marine Geodesy* 32(4):355–71.
- Brinkman, Richard Michael. 2006. "Wave Attenuation in Mangrove Forests: An Investigation through Field and Theoretical Studies." James Cook University.
- Camenen, Benoît, and Magnus Larson. 2007. "Predictive Formulas for Breaker Depth Index and Breaker Type." *Journal of Coastal Research* 23(4 (234)):1028–41.
- Cole, Thomas G., Katherine C. Ewel, and Nora N. Devoe. 1999. "Structure of Mangrove Trees and Forests in Micronesia." *Forest Ecology and Management* 117(1–3):95–109.
- Horstman, E. M., C. M. Dohmen-Janssen, P. M. F. Narra, N. J. F. van den Berg, M. Siemerink, and S. J. M. H. Hulscher. 2014. "Wave Attenuation in Mangroves: A Quantitative Approach to Field Observations." *Coastal Engineering* 94:47–62.
- Jadhav, Ranjit S., Qin Chen, and Jane M. Smith. 2013. "Spectral Distribution of Wave Energy Dissipation by Salt Marsh Vegetation." *Coastal Engineering* 77:99–107.
- Kirwan, Matthew L., and J. Patrick Megonigal. 2013. "Tidal Wetland Stability in the Face of Human Impacts and Sea-Level Rise." *Nature* 504(7478):53–60.
- Krauss, K. W., J. A. Allen, and D. R. Cahoon. 2003. "Differential Rates of Vertical Accretion and Elevation Change among Aerial Root Types in Micronesian Mangrove Forests." *Estuarine, Coastal and Shelf Science* 56(2):251–59.
- Lenk, Stephan, Diego Rybski, Oliver Heidrich, Richard J. Dawson, and Jürgen P. Kropp. 2017. "Costs of Sea Dikes – Regressions and Uncertainty Estimates." *Natural Hazards and Earth System Sciences* 17(5):765–79.
- Mazda, Yoshihiro, Eric Wolanski, Brian King, Akira Sase, Daisuke Ohtsuka, and Michimasa Magi. 1997. "Drag Force Due to Vegetation in Mangrove Swamps." *Mangroves and Salt Marshes* 1(3):193–99.
- Menéndez, Pelayo, Iñigo J. Losada, Saul Torres-Ortega, Siddharth Narayan, and Michael W. Beck. 2020. "The Global Flood Protection Benefits of Mangroves." *Scientific Reports* 10(1):1–11.
- Möller, I. 2006. "Quantifying Saltmarsh Vegetation and Its Effect on Wave Height Dissipation : Results from a UK East Coast Saltmarsh." *Estuarine, Coastal and Shelf Science*

- 69:337–51.
- Möller, Iris, Matthias Kudella, Franziska Rupprecht, Tom Spencer, Maike Paul, Bregje K. Van Wesenbeeck, Guido Wolters, Kai Jensen, Tjeerd J. Bouma, Martin Miranda-lange, and Stefan Schimmels. 2014. “Wave Attenuation over Coastal Salt Marshes under Storm Surge Conditions.” *Nature Geoscience* 7(September):727–32.
- Muis, Sanne, Martin Verlaan, Hessel C. Winsemius, Jeroen C. J. H. Aerts, and Philip J. Ward. 2016. “A Global Reanalysis of Storm Surges and Extreme Sea Levels.” *Nature Communications* 7(May).
- Narayan, S. ., T. .. Suzuki, M. J. Stive, H. J. Verhagen, W. Ursem, and R. Ranasinghe. 2011. “The Effectiveness of Mangroves in Attenuating Cyclone-Induced Waves.” *Coastal Engineering Proceedings* 1.
- Nicholls, Robert J., Daniel Lincke, Jochen Hinkel, Sally Brown, Athanasios T. Vafeidis, Benoit Meysignac, Susan E. Hanson, Jan-Ludolf Merkens, and Jiayi Fang. 2021. “A Global Analysis of Subsidence, Relative Sea-Level Change and Coastal Flood Exposure.” *Nature Climate Change* 11(4):338–42.
- Schuerch, Mark, Tom Spencer, Stijn Temmerman, Matthew L. Kirwan, Claudia Wolff, Daniel Lincke, Chris J. McOwen, Mark D. Pickering, Ruth Reef, Athanasios T. Vafeidis, Jochen Hinkel, Robert J. Nicholls, and Sally Brown. 2018. “Future Response of Global Coastal Wetlands to Sea-Level Rise.” *Nature* 561(7722):231–34.
- Vousdoukas, Michalis I., Lorenzo Mentaschi, Jochen Hinkel, Philip J. Ward, Ignazio Mongelli, Juan-Carlos Ciscar, and Luc Feyen. 2020. “Economic Motivation for Raising Coastal Flood Defenses in Europe.” *Nature Communications* 11(2119).
- Vousdoukas, Michalis I., Lorenzo Mentaschi, Evangelos Voukouvalas, Alessandra Bianchi, Francesco Dottori, and Luc Feyen. 2018. “Climatic and Socioeconomic Controls of Future Coastal Flood Risk in Europe.” *Nature Climate Change* 8(9):776–80.
- Vuik, V., S. N. Jonkman, B. W. Borsje, and T. Suzuki. 2016. “Nature-Based Flood Protection: The Efficiency of Vegetated Foreshores for Reducing Wave Loads on Coastal Dikes.” *Coastal Engineering* 116:42–56.
- Ward, Philip J., Brenden Jongman, Jeroen C. J. H. Aerts, Paul D. Bates, Wouter J. W. Botzen, Andres Diaz Loaiza, Stephane Hallegatte, Jarl M. Kind, Jaap Kwadijk, Paolo Scussolini, and Hessel C. Winsemius. 2017. “A Global Framework for Future Costs and Benefits of River-Flood Protection in Urban Areas.” *Nature Climate Change* 7(9):642–46.
- Willemsen, P. W. J. M., E. M. Horstman, B. W. Borsje, D. A. Friess, and C. M. Dohmen-Janssen. 2016. “Sensitivity of the Sediment Trapping Capacity of an Estuarine Mangrove Forest.” *Geomorphology* 273:189–201.
- Woodroffe, C. D., K. Rogers, K. L. McKee, C. E. Lovelock, I. A. Mendelsohn, and N. Saintilan. 2016. “Mangrove Sedimentation and Response to Relative Sea-Level Rise.” *Annual Review of Marine Science* 8(1):243–66.
- Yamazaki, Dai, Daiki Ikeshima, Ryunosuke Tawatari, Tomohiro Yamaguchi, Fiachra O’loughlin, Jeffery C. Neal, Christopher C. Sampson, Shinjiro Kanae, and Paul D. Bates.

2017. "A High-Accuracy Map of Global Terrain Elevations." *Geophysical Research Letters* 44(11):5844–53.

Zhang, Xiaofeng, Vivien P. Chua, and Hin Fatt Cheong. 2015. "Hydrodynamics in Mangrove Prop Roots and Their Physical Properties." *Journal of Hydro-Environment Research* 9(2):281–94.

Reviewer #2

General comments:

Reviewer #2 (Remarks to the Author):

In their paper entitled "Cutting the costs of coastal protection: how vegetation reduces global flood hazard," van Zelst and colleagues model the coastal hazard mitigation provided by mangroves and marshes on a global scale. They find that nearly a third of the global coastline is covered with vegetation and that this vegetation considerably lowers required levee height resulting in large cost savings. The analysis is based on global process-based models of wave evolution and new higher resolutions intertidal elevation datasets that make this wave modeling possible. The paper is an important contribution to the field -- specialists will be interested in reading it -- but I do have some reservations about publication in Nature Climate Change in its current state.

Reply:

Reviewer 2 expressed his/her interest in the manuscript.

The reviewer specifically highlighted the need to position this work in respect to other flood risk and ecosystem services literature, the need to elaborate on the feasibility of hybrid solutions in the long run and to acknowledge the downsides of the replacement costs approach.

We like to thank the reviewer for his/her kind words and his/her constructive feedback. Based on the review we:

- We elevated the novelty of the study
- We addressed the effect of sea-level-rise and long-term geomorphological change on the feasibility of hybrid solutions
- We extended the analysis by including population density and thereby revising the coastal protection needs assumption which makes the replacement costs approach more robust.

Hereafter a copy of the review by Reviewer #2 and a point-to-point reply on the raised issues.

Question 1:

1) The abstract states that the contribution of ecosystems to coastal flood risk has never been quantified at a global scale. This is misleading and reflects that the authors flood risk modeling perspective. They may be less familiar with the ecosystem services literature.

Beck et al 2018 Nature Communications quantify flood risk reduction provided by coral reefs globally using a similar modeling approach to this paper. Chaplin-Kramer et al. 2019 Science use a coastal hazard index to quantify the coastal risk reduction provided by corals, marshes, mangroves, and other coastal forests, as well as two other ecosystem services. Several studies

including and following on Costanza et al. 1997 use a benefit transfer approach to quantify the value of coastal protection services at a global scale. And Hochard et al 2019 use an empirical, statistical approach.

What is novel about this work is that it employs a process-based model for quantifying wave attenuation provided by vegetation, leveraging the newly available elevation data and accounting for spatial variation in key biophysical and economic variables. So it's more mechanistic than previous studies, which is important for understanding factors influencing mitigation and where to direct investments in restoration and conservation. I suggest the authors put the advancements of their study in the context of the previous ES literature as well as any relevant flood risk reduction papers.

Reply:

We are aware of the ecosystem services literature and admit that the position of the submitted manuscript in relation to the ecosystem services literature was not sufficiently put in context. An aspect that makes the current study novel is indeed the process-based approach and the consideration of both mangroves and salt marshes, as highlighted by the reviewer. What makes the manuscript unique is that it bridges two fields (flood risk and ecosystem services), because the manuscript focusses on hybrid solutions (combination of vegetated foreshore with a levee). The vegetated foreshore attenuates incoming waves and the levee defends against high waters. This design uses the strength of both modules. A solution solely focusing on mangrove green belt are often capable of attenuating incoming waves, but fail to prevent flooding as high water (surges) are hardly affected in absence of very extensive wetland's width. At the other hand, a purely 'grey' solution (a traditional levee or flood wall) is very costly if it has to withstand incoming waves at full force, but becomes financially attractive if combined with wetlands that attenuate these incoming waves. In addition, these hybrid solutions have the same co-benefits as full 'green' infrastructure, e.g. carbon storage, sediment trapping. We highlighted the focus on hybrid solutions in the revised manuscript (l. 39-49).

Question 2:

Another minor point is that the ecosystem services literature has few examples of urban ecosystem service analyses of coastal protection. This analysis explicitly breaks down urban and rural areas so it could in theory assess mitigation provided by vegetation for urban areas and elevate that point.

Reply:

In order to improve socioeconomic relevance, we extended the current analysis with the addition of population density results (Methods l. 156-160). Figure 3 in the revised manuscript gives insight in the distribution of the total levee cost reduction for multiple population density bins.

Question 3:

2) One issue with their approach to quantifying the reduction in costs is that it assumes levees at the end of the vegetation. It may be worth discussing limitations to this climate adaptation strategy. Would levees prevent the mangroves from migrating as sea-levels rise? What about sources of sediment to allow for accumulation?

Reply:

In global studies taking reasonable assumptions is inevitable. In this analysis we assumed the levee position at the end of the vegetated zone. Modelling of the faith of intertidal areas is depending on a complex interplay between tides, waves, wind and in some cases fresh water run-off, hampering long-term modeling of the morphological effects of accelerated SLR (Wang et al. 2018). Within green belts, vegetation contributes to stabilization of coastlines (Brinkman 2006; Horstman et al. 2014; Mcivor et al. 2013) and are to a certain extent able to grow with sea level rise (Lovelock et al. 2015). Although outside the scope of the current analysis, we are aware of the effects SLR on the functioning of hybrid coastal defense systems. In the revised manuscript this matter is addressed in the Discussion (l. 254-267 and 284-288).

Question 4:

3) Another issue that merits a least a couple of sentences is the replacement cost approach the authors use to quantify mitigation benefits. See Barbier et al. 2015 Ecosystem Services for a discussion of the limitations of the replacement cost approach. Economists generally think the avoided damages approach is superior. The RC approach estimates a benefit as a cost, human built alternatives are rarely more cost effective, and the RC approach makes the assumption that the alternative WOULD be built when that might not in fact be the case, especially at the global level and given the costs in remote places. So there are drawbacks of the economic approach used in this paper that should be discussed.

Reply:

There are multiple methods to express mitigation benefits. While it might be true that the avoided damages approach is perceived as superior by most economist, it tends to direct high potential locations towards urbanized (areas) with high economic value. We deliberately chosen the replacement coast approach and calculated these costs for nine protection levels. In addition, in the revised manuscript we extended the analysis using population density data to improve social-economic relevance. This reduces the chance that levees are assumed at locations where it is very unlikely they will not be built. The choice if to protect or not to protect and to what protection standard can be best supported by outcomes of a local avoided damages study. We elevated this matter in the discussion in the revised manuscript (l. 279-280).

Question 5:

4) The paper uses the well-regarded model, XBeach for the wave attenuation analysis. But it doesn't report the equation in the model accounting for attenuation in vegetation. That should be in the methods since it's a key aspect of the mechanistic approach. Also the analysis uses a single value for key parameters like height of mangroves etc., thus it doesn't capture variation in the attributes of vegetation even though it is capturing spatial coverage. This variation could have importance consequences for cost reduction provided by vegetation and is a limitation that should be mentioned.

Reply:

The equations in the used numerical model XBeach accounting for attenuation by vegetation are available in the referenced sources (van Rooijen et al. 2015; Van Rooijen et al. 2016). The equations that account for wave attenuation by vegetation are cited in the revised manuscript (Dalrymple, Kirby, and Hwang 1984; Mendez and Losada 2004), (Methods I. 138-140). In addition, the source code of XBeach is open and referenced as well (see code availability statement).

The current study considers indeed spatial coverage of vegetation, but applies a single value for tree height and marsh height. In future studies we like to introduce variations of these heights as well by coupling to NDVI measurements or other sources. For now, we applied single (conservative) factors (Methods I. 115-132). For example, the Methods mentions that we use young mangroves that are small and vertically uniform to prevent an overestimation of attenuation.

Question 6:

5) Legend in Fig 3 is confusing. What is meant by "high potential countries"? I follow the logic of including the percentage of population exposed in the risk reduction score per country, but I would also point out that the authors are mixing a risk approach (exposure/consequence) with a quantification of costs/levee height reduction approach. Did the authors use any theory / conceptual model to base this on? The conceptual model and analysis for the biophysical part of the modeling is strong, but the links between the biophysical and economic/social side and the conceptual model for economic analysis are weaker.

Reply:

We disregarded Figure 3 as present in the initial submission. Instead we constructed a new figure that gives more insight in the sensitivity of the results on population density, levee construction costs, protection standards and differences between mangroves and salt marshes.

Question 7:

6) In some places there seems to be more jargon (e.g., “required crest heights for 100-yr protection standard” than appropriate for an interdisciplinary journal such as Nature and the results are very detailed with a lot of % rather than compelling descriptions of the magnitude of differences in words. This struck me as also less appropriate for a high impact journal such as Nature CC. In addition, I found some colloquial language such as “profits will for sure result” on line 236, and lots of missing words or incoherent lines that suggest the paper needs a good proof read—even in the last sentence of the abstract.

Reply:

In the revised manuscript we adapted the Introduction and especially the Discussion such that these sections are more appropriate for an interdisciplinary audience. We kept the wording in the Results section to the point and factual to ensure rigidity of the manuscript. We gave the manuscript another good thorough read. We hope the language in revised manuscript satisfies the reviewer’s expectations and are very willing to apply further corrections if required.

Question 8:

The authors start to make some important points in the paragraphs starting on lines 195 and 206, but these paragraphs are unclear and sentences don’t necessarily segway from each other. For instance they say these ecosystems also provide other services – why is that important to this study? Grey infrastructure cannot provide ES – would info on those services inform the calculus further?

Reply:

We inserted the statement on other ecosystem services to inform readers not familiar with this topic. We acknowledge that the statement might be poorly positioned and is moved to the introduction (l. 31-33) in the revised manuscript. We choose not to delete the statement, as another Reviewer even suggested that we should elaborate on all ecosystem services provided by mangroves and salt marshes. We tried to satisfy both reviews by keeping this statement in the manuscript, but not elaborating further on the other services.

Question 9:

7) The authors may be interested in Silver et al. 2019 Frontiers in Marine Science as an example of a recent coastal protection analysis in a SIDs with long rural coastlines not likely to be protected with infrastructure (see first paragraph of the discussion).

Reply:

We incorporated the suggested literature into the revised manuscript (l. 141-142).

Yours sincerely,

Vincent T. M. van Zelst ^a, Jasper T. Dijkstra ^a, Bregje K. van Wesenbeeck ^{a, b}, Dirk Eilander ^{a, c},
Edward P. Morris ^{d, e}, Hessel C. Winsemius ^{a, b}, Philip J. Ward ^c and Mindert B. de Vries ^a

^a *Deltares, P.O. Box 177, 2600 MH Delft, The Netherlands*

^b *Delft University of Technology, Faculty of Civil Engineering and Geosciences, P.O. Box 5048, 2600 GA Delft, The Netherlands*

^c *Institute for Environmental Studies (IVM), Vrije Universiteit Amsterdam, 1081 HV Amsterdam, The Netherlands*

^d *Instituto Universitario de Investigación Marina (INMAR), University of Cádiz, 11510 Puerto Real, Cádiz, Spain*

^e *EO4 Data Science, 11500 El Puerto de Santa Maria, Cadiz, Spain*

References reply reviewer #2

- Brinkman, Richard Michael. 2006. "Wave Attenuation in Mangrove Forests: An Investigation through Field and Theoretical Studies." James Cook University.
- Dalrymple, R. A., J. T. Kirby, and P. A. Hwang. 1984. "Wave Diffraction Due To Areas of Energy-Dissipation." *Journal of Waterway Port Coastal and Ocean Engineering-Asce* 110(1):67–79.
- Horstman, E. M., C. M. Dohmen-Janssen, P. M. F. Narra, N. J. F. van den Berg, M. Siemerink, and S. J. M. H. Hulscher. 2014. "Wave Attenuation in Mangroves: A Quantitative Approach to Field Observations." *Coastal Engineering* 94:47–62.
- Lovelock, Catherine E., Donald R. Cahoon, Daniel A. Friess, Glenn R. Guntenspergen, Ken W. Krauss, Ruth Reef, Kerry Lee Rogers, Megan L. Saunders, Frida Sidik, Andrew Swales, Neil Saintilan, Le Xuan Thuyen, and Tran Triet. 2015. "The Vulnerability of Indo-Pacific Mangrove Forests to Sea-Level Rise." *Nature* 526(7574):559–63.
- Mcivor, Anna, Tom Spencer, Iris Möller, and Mark D. Spalding. 2013. "The Response of Mangrove Soil Surface Elevation to Sea Level Rise Natural Coastal Protection Series: Report 3." *Natural Coastal Protection Series ISSN 2050–7941*.
- Mendez, Fernando J., and Inigo J. Losada. 2004. "An Empirical Model to Estimate the Propagation of Random Breaking and Nonbreaking Waves over Vegetation Fields." *Coastal Engineering* 51(2):103–18.
- Van Rooijen, A. A., R. T. McCall, J. S. M. Van Thiel De Vries, A. R. Van Dongeren, A. J. H. M. Reniers, and J. A. Roelvink. 2016. "Modeling the Effect of Wave-Vegetation Interaction on Wave Setup." *Journal of Geophysical Research: Oceans* 121:4341–59.
- van Rooijen, A. A., J. S. M. van Thiel de Vries, R. T. McCall, A. R. van Dongeren, J. A. Roelvink, and A. J. H. M. Reniers. 2015. "Modeling of Wave Attenuation by Vegetation with XBeach." *E-Proceedings of the 36th IAHR World Congress* 7.
- Wang, Zheng Bing, Edwin P. L. Elias, Ad J. F. Van Der Spek, and Quirijn J. Lodder. 2018. "Sediment Budget and Morphological Development of the Dutch Wadden Sea: Impact of Accelerated Sea-Level Rise and Subsidence until 2100." *Geologie En Mijnbouw/Netherlands Journal of Geosciences* 97(3):183–214.

Reviewer #3

Reviewer #3 (Remarks to the Author):

General comments:

Key points:

The paper addresses an important topic in light of recent, growing attention given to 'nature-based' solutions towards coastal flood protection and mounting evidence that vegetated foreshores are of critical importance in buffering against the impact of extreme storm/wave events. However, two key points have led me to recommend rejection of this paper.

Reply:

Reviewer 3 initially recommended to reject the manuscript.

The reviewer specifically highlighted methodological issues that come with global assessments and conceptual issues ranging from not including other ecosystem services to lacking amount of detail and explanation in the main manuscript.

We like to thank the reviewer for his/her kind words and his/her constructive feedback. Based on the review we:

- We elevated the usefulness of global studies and underlined the issues that should be considered in local evaluations.
- We did not transfer details from the methods sections in the main manuscript as we do not think this is appropriate for the interdisciplinary audience.
- We extended the study with a sensitivity analysis

Hereafter a copy of the review by Reviewer #3 and a point to point reply on the raised issues.

Question 1:

1) Methodological issues: the expectation of the authors to be able to estimate this very locally determined function of vegetated coastal ecosystems is fundamentally flawed. Waves reaching the upper intertidal zone are the result of morphodynamic interactions that begin a considerable distance seaward/below the limit of the vegetation present in the intertidal zone, particularly on shallow sloping coasts, where waves begin to be dissipated several kilometers before they reach the upper intertidal zone, such that plants can establish there. Without confidence in accurately predicting not just the intertidal topography but also the sub-tidal, nearshore topography, the application of the methods reported here is, to my mind, fundamentally problematic (at least there is no concrete evidence presented in the paper to convince the reader that the results are robust and errors are not reported for the global analysis).

Reply:

The method should be in line with the scope of the study. Global studies cannot and do not aim to predict interactions with a similar accuracy as local studies, but use reasonable assumptions to be able to highlight the potential and locations of the studied matter. Global assessment have their limitations and a mismatch between model abilities and expectations of users should be addressed (Ward et al. 2015). The current study is meant to identify areas where it matters most and to stimulate further detailed local assessments in which all interactions should be considered explicitly. We further underlined this in the discussion of the revised manuscript (l. 275-288).

In a global assessment, interactions that can be well addressed in local studies are impossible to simulate with a similar level of detail without very intensive modelling and a good set of validation data. Especially the latter is problematic as field data on wave attenuation by mangroves and salt marshes during extreme conditions does not exist. However, the goal is not to simulate wave-bed interaction to the same level of detail on a global scale compared to a local study. For a global assessment not explicitly simulating wave-bottom interaction in the sub-tidal is not wrong, as long as wave dissipation that takes place before the intertidal zone is taking into account. In this study, we assume that waves at the start of the foreshore are depth limited and thus determined by the wave height to depth ratio. We applied a low conservative breaking index of 0.55 and thereby implicitly accounting for processes that take place outside the intertidal zone.

Question 2:

Figure 6 to 8 are critical here in that they illustrate the wide spread of the data around the best-fit line between modelled and validation data. With respect to vegetation width, e.g., it is not uncommon for widths to be over or under-estimated by 50-100% which is a significant error term. The authors only explicitly state what the overall errors were (absolute and relative) in wave dissipation estimates for a particular location (Lines 244-251) but then report global estimates.

Reply:

Global datasets and assessment based on those kinds of datasets have larger error margins. The goal of the study is not to accurately predict wave-vegetation interaction on all individual locations around the globe, but to estimate the overall potential and highlight areas of interest. To support this target it is important to aim for a spread around the best-fit line without a strong bias, especially not a bias to overestimate overall potential. To achieve this, we choose conservative factors for various parameters such as vegetation characterization and the breaking index.

We performed an additional sensitivity analysis to give insight in the effect of over or underestimated factors, such as vegetation width as highlighted by the reviewer. For the vegetation width we used the 75% confidence interval from the vegetation width validation as input (Methods I. 260-282). The results indicate that in case of a structural underestimation/overestimation of the vegetation width that the levee costs reduction potential would be 28 % / 39 % lower / higher. This might seem as a larger figure, but in global assessment uncertainties in the natural system (vegetation characterization, hydrodynamics) are often subordinate to uncertainties in the socio-economic system. The main study result, the levee cost reduction potential, is the most influenced by the levee costs and the assumptions on coastal protection needs. We introduced a new figure (Figure 3 in the revised manuscript) to give clear insight in the influence of socio-economic data (for this study specifically population density in areas prone to coastal flooding, protection level and levee construction costs) on the overall results.

Another aspect is the lack of availability of calibration data on wave-vegetation interaction during storm conditions. Upscaling models with settings calibrated on daily field conditions and laboratory experiments is to date the best option combined with validation of data layers that are used in between steps. Putting a lot of effort in fine-tuning the model to improve vegetation determination and wave-vegetation interaction capabilities might give a false sense of accuracy given the large spread in the socio-economic data and it will not change the overall message and hotspots communicated in the manuscript.

Question 3:

They do not state what the expected accuracy is in the height estimates at the seaward margin of the vegetation globally, nor how coastal bathymetry in the sub-tidal zone is taken into account (where waves are already affected by the bed). Without this, the dissipation percentages reported here at the global scale rather lose their value (nearshore bathymetry arguably changes over similar time-scales, if not more rapidly, than the upper intertidal shore (whether through human intervention or 'naturally')). I rather suspect that errors in bathymetry and closer to shore away from the test sites can be many times larger than at the 'test-location' reported here on many transects globally, due to the sensitivity of wave conditions to nearshore (as well as intertidal) elevations. Where waves are likely to be depth limited, even the reported accuracy of 50cm in the intertidal topography is likely to be significant in terms of its impact on wave height and wave dissipation estimates.

Reply:

The accuracy of the bathymetric and topographic data layers are described in the Methods-elevation data section (Methods I. 9- 34). We acknowledge the importance for bathymetry and elevation data with high accuracy. Especially for this reason, we produced specifically for this study the intertidal topography layer with an accuracy of 50 cm. The complementary data layers

(GEBCO and MERIT) have a lower accuracy, but are commonly used in global studies such as the *Nature* publications by (Athanasίου et al. 2020; Menéndez et al. 2020; Muis et al. 2016; Vousdoukas et al. 2018, 2020). The inclusion of geomorphological changes is out of the scope of this study, because to our knowledge no accurate time-varying sub-tidal/intertidal dataset exists. Taking into account these processes by explicitly modelling morphological change (and even vegetation growth) is to date only possible on a local scale due to data requirements and computational efforts. However, we looked into the impact of the reported accuracy of 50 cm onto levee cost reduction results. This additional analysis indicates that a shallower/deeper topography would yield a -39 %/ 47% change in levee cost reduction potential due to vegetation presence (Supplementary Fig. 10). The levee cost reduction potential is influenced most by assumptions on coastal protection needs (protection level, people exposed) and the levee construction costs (l. 208-216).

Question 4:

2) Conceptual issues: The use of the term 'ecosystem' in the context of wave dissipation can be a bit misleading – it is really the landforms that are associated with the ecosystems here that fulfil much of the protecting function as they constitute an altered bathymetry in the nearshore.

Reply:

We do recognize that ecosystem service wave dissipation can be prone to ambiguity, as there are multiple mechanisms that contribute to wave dissipation. In this study we focus solely on wave-vegetation interaction (interaction between waves and the mangroves and salt marshes) as stated multiple times in the main manuscript. We used a conservative approach in which the landform for the situation with and without vegetation presence remains the same. We acknowledge that by trapping of sediment and other mechanisms the landform is established and stabilized by the presence of the vegetation and that this landform can contribute significantly to wave dissipation. However, (Willemssen et al. 2020) shows that especially during extreme surge levels the ratio of wave-attenuation by vegetation over wave energy dissipation on the tidal flat increases. In our analysis, the attenuation results represent wave energy dissipation due to wave-vegetation only (l. 111-113). Energy dissipation rates along the vegetated foreshore, including the effect of bottom friction and depth-induced breaking on tidal flats, will logically exceed the presented numbers.

Question 5:

The results are presented with very little detailed information on context. See, for example, my comment on line 83. Presumably, the authors have determined these percentages of wetland fringed coasts and flood susceptible coasts on the basis of the 1km spaced normal coastal transects?

Reply:

The study results are indeed determined based on normal coastal transects. This information was written in the initial manuscript on lines 63 (revised manuscript l. 82-83). Additional information is available in the Methods section (Methods l. 2—8). We choose to place these details into the Methods to improve overall readability for a multidisciplinary group.

Question 6:

Flood risk, particularly due to wave action, can be much more locally variable than this (as it may depend on abrupt changes to coastal landform configuration, e.g. breaching etc..). The authors should at least acknowledge this, if not comment on the implication of ignoring this.

Reply:

We like to underline again that the target of this study is not to calculate flood risk on such a local scale. Nevertheless, in the Discussion of the revised manuscript we acknowledged that processes related to coastal landform configuration cannot be fully captured with a 1-dimensional transect approach (l. 280-282).

Question 7:

Some of the results statements appear entirely unsupported, e.g. lines 84-85: 'These vegetated foreshores on susceptible coastlines are encountered to a similar extent at both urban and rural coastlines' – no evidence is presented to back up this statement. What criteria was applied to classify a coast as 'urban' versus 'rural'?

Reply:

The classification of urban and rural coastlines is described in the Methods section under the header '*Coastline susceptible to flooding, urban and rural extents and population density*' (Methods l. 147). In the main manuscript we changed the addressed lines by inserting the actual vegetation percentages along the rural and urban populated coastlines susceptible to flooding (l. 103-104).

Question 8:

The discussion section makes many claims (e.g. about the benefit of coastal vegetation in the first sentence and the claim that small island development countries largely depend on wave reducing abilities of coastal vegetation) without referring to relevant literature (there is an almost complete lack of citation) or results provided in the paper itself.

Reply:

The Discussion in the revised manuscript has been updated with additional literature references and references to results in the manuscript itself.

Line-by-Line Comments:

Question 9:

Line 12: 'growing population and economy' sounds a bit strange... The economy is not necessarily growing everywhere... Do the authors really mean growing 'economic activity'?

Reply:

Here, we applied the exposure definition as published by the United Nations International Strategy for Disaster Reduction. It defines exposure as people, property, systems or other elements present in hazard zones that are thereby subject to potential losses. We are not stating that economy is necessarily growing everywhere, but that globally exposure to coastal floods is increasing due to a growing economy and population.

Question 10:

Line 20: '... financially attractive designs.' – of what? It would help if the authors were a bit more explicit here.

Reply:

Financially attractive coastal protection strategies. We changed the sentence for increase clarity (l. 19).

Question 11:

Line 20-22: 'We calculated that...' and also Line 75: 'For areas with no levees....' – to present this suggestion rather uncritically here can be misunderstood. Are the authors really contemplating the establishment of levees along all coastlines susceptible to flooding? That is clearly a very short-sighted idea as there are many coastlines susceptible to flooding where it would be entirely foolish to build any kind of levee – as it is well known that the construction of levees encourages settlement and development landward of the levees and in fact may increase the population and asset value at risk from flooding, even though it reduces the likelihood of flooding over the short term.

Reply:

There are various ways to manage flood risk, e.g. hard structural measures such as flood walls, but also soft solutions, hybrid solutions or non-structural solutions like early warning systems. We are very well familiar with the levee effect, but also with the fact that over 600 million people are currently at risk (McGranahan, Balk, and Anderson 2007) and that decision-makers and coastal planners are looking for action perspective. We do not state that levees should be constructed along all populated coastlines that are susceptible to flooding. We calculated the potential savings to exemplify the potential of combining existing vegetation with levees. We considered applying a threshold in terms of population and assets to determine if construction

of levees would be a reasonable option. However, every country has its own vision and policy on coastal protection needs. We concluded that as scientist we want to remain independent in this matter and highlight the overall potential.

Question 12:

Even intertidal ecosystems will eventually cease to protect the levee, where sea level rise and lack of sediment delivery constrain the future existence of the ecosystem....

Reply:

This is valid long-term concern that we elevated in the discussion of the revised manuscript, as it is important to make people aware of this matter. In the Discussion in the revised manuscript we reflected more extensively on effects of SLR (l. 254-267). However, we are confident that with a multidisciplinary approach sustainable hybrid coastal protection strategies can be designed and implemented.

Question 13:

Line 30: 'purely grey solution' – the 'grey' should be in inverted commas.

Reply:

Incorporated in the revised manuscript (l. 30).

Question 14:

Line 31: refers to 'ecosystem services' but the text before this only identifies ONE ecosystem service. The authors should elaborate on the many other ecosystem services provided by coastal ecosystems.

Reply:

There is some disagreement between the reviewers whether to elaborate on other ecosystem services or not. As middle ground we mentioned a few other ecosystem services with references to literature for further reading in the revised manuscript (l. 31-33).

Question 15:

Line 31: the sentence starting 'Despite...' sounds as though providing ES should be protecting wetland areas from decreasing... Please rewrite.

Reply:

It is ironically that these valuable ecosystem are decreasing given the broad set of services they provide. In the revised manuscript we rewrote this sentence (l. 33-34).

Question 16:

Line 32: I think it should be 'Over the short term...' and 'over the long-term...' and the authors should give an indication of what they mean by either. The point about 'loss of coastal lands' is a little naïve – as the loss of coastal land is part of naturally dynamic coasts that erode in some places and form new landforms in others. There is no recognition in here of this.

Reply:

We checked the correctness of the used wording in the Cambridge dictionary. In the revised manuscript we inserted an indication of what we mean with 'long term' and 'short term'. In addition, we recognized that coastlines are inherently dynamic (l. 34).

Question 17:

Line 49: The authors state that '...the effect of coastal wetlands on storm surge levels is typically only relevant for very large wetlands and still leads to flooding in areas that are not protected by levees'. This statement seems confusing without any further explanation here. Are the authors saying that coastal wetlands can exacerbate flooding in areas that are not protected? Or are they simply saying that coastal wetlands reduce storm surge levels but not where there are no wetlands (which seem rather obvious...)? This needs rewriting/clarifying.

Reply:

We rewrote this part in the revised manuscript (l. 68-71). In the study we omitted the effect of wetlands on storm surge levels, because it only becomes relevant for very extensive wetlands of multiple kilometres and largely depends on the local spatial configuration that is not captured with 1D transects (see also the Discussion section). We inserted additional literature to back-up these statements.

Question 18:

Line 52: What is meant by 'adaptive designs' here? Adaptive to what precisely? There are many changes that one might want to adapt coastal protection to...

Reply:

Nature-based Solutions are typically more flexible compared to traditional coastal protection infrastructure, due to their inherent dynamic character they can be (to a certain extent) adapted to new information and information that may emerge. The type of information can be numerous, e.g. new information on rising sea levels, but also for example new regulations on coastal protection levels. We rewrote this sentence for clarification.

Question 19:

Line 54/55: This is a good example of where the authors are using a rather unclear and unhelpful written style: '... at the end of the vegetated foreshore' – where would the 'end' of a

'vegetated foreshore' be? Is it the Highest Astronomical Tide level? With or without extreme meteorological forcing?

Reply:

Profile construction is described in the Methods section (Methods I. 66-83), as we argue this is too detailed for the main text. The end of the vegetated zone is defined as the last profile point that is vegetated with mangroves or salt marshes. This can be at the Highest Astronomical Tide, but could also be lower or higher in the profile depending on vegetation presence in the FAST vegetation presence map, the salt marsh map (Mcowen et al. 2017) and the mangroves map (Giri et al. 2011).

Question 20:

Line 59: What are the 'existing vegetation maps' referred to here?

Reply:

The 'existing vegetation maps' refer here to the vegetation cover maps from GlobCover, CorineLandCover, salt marsh map (Mcowen et al. 2017) and mangroves map (Giri et al. 2011). These sources were referenced in the Methods, but are now also referenced in the main text (l. 78).

Question 21:

Line 62/63: surely the 'polar circle' is either North or South pole, so the 'and -60 S' is not clear here.

Reply:

In the revised manuscript we inserted the actual degrees latitude and removed 'polar circle'. The study domain is between 66 degrees North and 60 degrees South (l. 82).

Question 22:

Line 61: The ERA-Interim data ocean wave model horizontal resolution is 110 km with wave spectra discretised using 24 directions and 30 frequencies – with many coastal wetland coasts extending in length for less than this distance and often situated in topographically complex embayments or estuaries, it is not clear here how the authors

Question 22.1:

(a) interpolated to finer resolution to capture the effect of such wetlands on such coasts and

Reply:

The ERA-Interim grid consists of 241 (latitude) by 480 (longitude) points (resolution 0.75°). We projected these gridded data points onto the Dynamic Interactive Vulnerability Assessment

(DIVA) points (16.394 points) using a nearest function. Next, we calculated significant wave height and the peak wave period per DIVA point using a Peaks Over Threshold (POT) method using the ORCA-toolbox (<https://www.deltares.nl/en/software/orca/>).

Question 22.2:

(b) took account of wave refraction/reorientation over complex bathymetry – something that may lead to wave dissipation (or at least along-shore redistribution of wave energy) from the offshore to the nearshore regions in its own right.

Reply:

Wave-bottom interactions such as refraction are not explicitly simulated. We assumed the presence of depth limited waves at the start of the intertidal zone using a breaker criterion of 0.55. Explicitly modelling spatial redistribution of wave energy over complex bathymetry is on a global scale only possible using large computational efforts. In our view this exercise will not yield a meaningful outcome to the levee cost reduction results looking at the uncertainty in socio-economic aspects.

Question 23:

It is also unclear whether the models accounted for wind direction at the time of wave modelling. In the shallow environments in which coastal wetlands grow, waves are likely to be depth limited by the time they reach the wetland. They lose a significant amount of energy due to breaking in the process and whether winds are blowing on- or off-shore can make a significant difference to wave height. The authors do not comment on this.

Reply:

We modeled wave propagation assuming a coastal normal wave direction and excluded wind-induced wave growth along the transect. We inserted modifications in the Methods section 'Water level and wave data' to make these aspects clearer.

Question 24:

Line 67/68: In considering this comparison between the foreshore with and without vegetation, the authors need to consider that the landform that determines the bathymetry and on which the vegetation is growing is itself the result of the presence of the vegetation – i.e. vegetated coastal wetlands are formed by both externally derived (allochthonous) and internally derived (autochthonous) organic material through plant growth.

Reply:

We assume the same landform for the vegetated and unvegetated model simulations. This is in line with our approach that we consider the present state situation. Taking into account biotic

and abiotic processes that result in landform change is out of the scope of the current study. Assuming the same landform is a conservative approach, including land form alternation will likely increase the wetland potential to reduce levee costs. In the revised manuscript landform change receives additional attention in the Discussion, furthermore we underlined in the revised manuscript that the landform for the vegetated and unvegetated situation is assumed the same (l. 88).

Question 25:

Line 83: '...27% is susceptible to coastal flooding' – the authors do not say over what time period? How was this figure arrived at?

Reply:

The derivation of coastlines susceptible to coastal flooding is considered to detailed for the main text. The derivation method is described in the Methods section 'Coastline susceptible to flooding, urban and rural extents and population density' (Methods l. 144).

Question 26:

On line 95, the authors refer to the 'non-linear relation between wave attenuation and vegetation width'. In fact, the literature has shown that significant wave attenuation occurs over only a 40m distance of vegetated coastal wetlands and certainly over an 80m distance, after which little wave energy is left.

Reply:

The referee refers to statements in literature that 'significant' wave attenuation occurs over rather short distances. What literature is unclear, but observations confirm the ability of mangroves and salt marshes to reduce incoming waves during daily conditions. Our study focusses on wave attenuation by foreshore vegetation during storm events during which wave heights and water depths occur that exceed daily observations. McIvor et al. (2012) gives an overview of wave height reduction by mangroves and states that wave height can be reduced by 13 % and 66 % over 100 metres of mangroves. First, there seems to exist quite a range between the minima and maxima attenuation over this stretch, due to for example influence of water depths, topographies, tree morphology related to water depth and wave height. Möller et al. (2014) and Rupprecht et al. (2017) reports the findings of a full-scale flume test of natural salt marsh under storm conditions. Here, salt marshes attributed up to 60 % of wave reduction, but lower attenuation rates were observed for higher wave heights and different salt marsh types. These studies exemplify that in some instances significant wave attenuation can be accomplished in short distances, but that to several hundreds of metres are required to dissipate waves fully.

Question 27:

The authors state that 13.1% of coastal wetlands occupy a cross-shore width of 0-100m (figure 1A) but it seems that this class includes '0', i.e. it is not clear what percentage has a width that exceeds a width over which any degree of wave dissipation can be expected. How accurate the width estimates? What error margin is involved in these estimates?

Reply:

We understand the class minimum of '0' is confusing. Profiles are constructed using a horizontal grid resolution of 25 metres. Over this width wave dissipation can be expected. In the revised manuscript we changed the lower bound '0' to '25' meters in Figure 1a.

Question 28:

Line 159: '... where vegetation protects most assets and people' – vegetation alone does not protect assets and people, thus this needs to be changed to 'vegetation helps to protect....'...

Reply:

Changed accordingly reviewer's suggestion in the revised manuscript (l. 188).

Question 29:

Line 161: sentence does not make grammatical sense.

Reply:

Rewritten in the revised manuscript (l. 190 -191).

Question 30:

Line 167-168: this mentions errors but does not give error bands or any concrete information on them. '...some results may be under- or overestimated' is simply far too vague for a scientifically credible paper and journal.

Reply:

We performed an additional sensitivity analysis (Methods l. 256-278) to give insight in the influence of uncertainty of various aspects of the study (e.g. bathymetry, hydrodynamic boundary conditions, levee construction costs). The results of this analysis are described in the Discussion of the revised manuscript (l. 206-213). A figure corresponding to this analysis is added (Supplementary Figure 10).

Question 31:

Line 171: what is meant by 'slightly underestimated'? See comment above – this needs to state the quantitative uncertainty estimates here to be credible. The same applies to any other reference to errors here (e.g. 'an underestimation of wave attenuation' – by how much? – then (on line 174) 'an overestimation of wave attenuation results' – by how much?). How credible are the full study results if there are significant under and over estimations reported?

Reply:

The degree of underestimation of waves and extreme water levels in ERA-I and GTSM are quantitatively included in the Discussion of the revised manuscript (l. 197-198).

METHODS:

Question 32:

Line 71-74: the authors state "SLR and subsidence were not taken into account because this study focuses on the present situation. Moreover, quantifying the future role of vegetated foreshores would not only require sea level rise scenarios but also an insight in the development of wetlands over time, which is strongly determined by local conditions such as sediment supply." –

but (1) no references are given to back up the latter statement and (2) these constraints are not highlighted sufficiently in the main body of the paper and its concluding section (thus not providing a sufficiently balanced and cautionary approach towards the full implications of the results of this study – something that is of utmost importance to a high-impact journal such as this).

Reply:

In the revised manuscript we inserted additional references to the Methods section (Methods I. 53). In addition, we mentioned this subject in the Discussion of the revised manuscript (l. 253-254 and l. 281-285).

Question 33:

Line 77: Here and in the results and discussion, the authors never state what the distribution of above-wetland wave heights was that this modelling resulted in. The breaking criterion and the fact that vegetated wetlands (marshes and mangroves) only exist at the very landward limits of the intertidal profile, results in relatively small wave heights at the seaward margin of the vegetated zone. It also means that the unvegetated surfaces seaward of the vegetated wetland are just as important at re-distributing and dissipating wave energy as, without such re-

distribution/dissipation, the wetland would most likely not (be able to) exist. The authors rather gloss over this in both discussion and methods.

Reply:

The average and standard deviation of the wave heights offshore and at the start of the foreshore are included in the Methods (Methods I. 65). The average wave heights at the end of the (vegetated) foreshores is also included (Methods I. 143). The figure below shows the distribution of the wave heights at various locations along the coastal normal transects. In our analysis, the significant wave height at the end of the vegetated foreshore is on average only half (difference = 30 cm on average) of the significant wave height in case of a model simulation of a bare foreshore.

Figure 2 Significant wave height distribution at various locations along the coastal normal transects (combined data for all nine return periods).

Question 34:

Line 82 onwards: Profile construction: The key issue here relates to my comment above: the authors frequently here make statements such as ‘the accuracy [...] required...’ or ‘validity checks were performed...’, or ‘a check was performed...’ without giving detail on what the exact criteria were for discarding/accepting data and what the implications for accuracy/error were overall as a result.

Reply:

In the revised manuscript we have made the thresholds that determine whether data is considered invalid and thus removed (more) clear (Methods I. 66 – 111).

Question 35:

What was the estimation/modelling error for water depth? What was the estimation/modelling error for wave heights along the transect? It is extremely important to communicate the relative magnitude of the estimated values alongside their errors. The authors only report on percentage wave height change but even this is not communicated clearly with error margins.

Reply:

It is difficult to address this comment, as a clear reference to the lines that the reviewer is referring to is missing. Between lines 82 and 128 of the Methods in the initial manuscript are no sentences that communicate modelling results. I assume the reviewer is interested in the average water depth error. This error constitutes from the vertical accuracy of the elevation data and the water level data that are communicated in the Methods sections 'Elevation data' and 'Water level and wave data'. Next, for Figure 3 in the revised manuscript we have included bars for wave attenuation representing the 85%-ile and 15%-ile. In the additional sensitivity analysis, uncertainty on topography and hydrodynamic conditions (based on different return periods) are addressed.

Question 36:

Line 128-130: The authors state "The modelling approach uses four parameters to represent vegetation: height, diameter, number of stems and drag coefficient. The exact characteristics are based on observations in literature (N =30 m⁻², d = 35 mm, h = 3.0 m)." – but again, no reference is made to where in the literature (what literature) these figures come from. Surely, these parameters vary very widely around the world for different wetland types at different stages of evolution/growth?!?

Reply:

Salt marsh characteristics are derived based on (Jadhav, Chen, and Smith 2013; Möller 2006; Möller et al. 2014; Vuik et al. 2016) and field observations at the FAST field test sites in Romania, UK, Spain and the Netherlands (<https://zenodo.org/record/581247#.XtkSljzbs0>). The latter provided data on seasonal variation of the leaf area index of salt marshes in temperate areas. The data showed a clear peak of the LAI in summer. We selected a parameterization representative for a winter state as found in NW Europe, which is a conservative assumption. The applied drag coefficient (required for numerical modelling of wave-vegetation interaction) was set to 0.19 based on large scale flume tests (Möller et al. 2014). Mangrove characteristics are derived based on field observations by (Brinkman 2006; Cole, Ewel, and Devoe 1999; Horstman et al. 2014; Krauss, Allen, and Cahoon 2003; Mazda et al. 1997; Narayan et al. 2011; Zhang, Chua, and Cheong 2015). From these sources we derived a characterization of (fringing) pioneering mangroves.

Salt marshes and mangroves indeed vary around the globe. For this reason, we chose a conservative characterization. Modelling of spatially varying vegetation characterization falls out

of the scope of the current research, because a global dataset of salt marsh species and mangroves species distribution along global coastlines does to our knowledge not exist.

Question 37:

Line 134: "A dragcoefficient (CD) of 0.19 is chosen, which is the lower limit found during large scale flume tests." – another statement with no reference to the literature and/or indication of how varied this parameter can be expected to be across the wetlands included in this study.

Reply:

In the revised manuscript we inserted the missing reference (Möller et al. 2014). In addition, we highlighted the factors that influence the choice for the drag coefficient in the Methods (Methods I. 125-126).

Question 38:

Line 160 onwards: the authors do not define what they mean by 'susceptibility to flooding' accurately enough. Is this just a definition by elevation of the coastal hinterland? Surely, it matters what the configuration of the coast is as a three-dimensional space and, as the authors themselves acknowledge earlier on, what is currently present between the intertidal vegetation and the landward areas that are 'susceptible to flooding'.... More information is needed here and a more critical / objective / comprehensive definition.

Reply:

The susceptible coastline is derived based on a GIS model that includes the three-dimensional space and also takes into account land surface roughness, which makes this method more sophisticated in comparison to a simpler 'bathtub' approach. In the revised manuscript we made this definition (more) clear. A more detailed description of this method can be found in the referenced literature (Haer et al. 2018).

Question 39:

Line 184 onwards: Reliability: the authors introduce the 'scoring classes' but do not define what the actual or RMS errors are that equate to these classes.

Reply:

In the section starting at l. 186 in the revised manuscript is described that the scoring classes for hydrodynamics, vegetation and profile elevation are based on a combined set of quantitative and qualitative indicators.

Question 40:

Line 209: This is the first time 'FAST TEA' is mentioned. What does it stand for?

Reply:

FAST stands for Foreshore Assessment using Space Technology and TEA stands for time-ensemble average, which is a technique that is used in development of the FAST intertidal elevation map. This technique is described in the Methods section 'Elevation data' (Methods I. 9-33).

Question 41:

The figures are generally helpful, although as stated above, it would have been useful to have the actual wave heights at the seaward margin of the vegetation presented in a figure to show how high incident waves were on these types of coasts prior to being dissipated - i.e. what magnitude of a problem is the paper addressing?

Reply:

We hope that the histograms in Figure 2 inserted in this document provide insight in this matter. The wave heights at the start of the foreshore could exceed one metre and our analysis underlines that wave-vegetation interaction attenuates a large share of the incoming nearshore wave energy.

We would like to thank the reviewer again for his/her extensive review. His/her efforts to fully assess our manuscript and write this review is very much appreciated. To conclude, we hope that all points of the reviewer are clarified to a satisfactory degree. While writing the manuscript we had to make inevitable decisions on the level of detail for the various aspects of the study given the boundaries of the maximum amount of words and the multidisciplinary character of the journal. We are confident that the revised manuscript is robust and provides more insight on the essential matters, while still being accessible for a broad readership.

Yours sincerely,

Vincent T. M. van Zelst ^a, Jasper T. Dijkstra ^a, Bregje K. van Wesenbeeck ^{a, b}, Dirk Eilander ^{a, c}, Edward P. Morris ^{d, e}, Hessel C. Winsemius ^{a, b}, Philip J. Ward ^c and Mindert B. de Vries ^a

^a Deltares, P.O. Box 177, 2600 MH Delft, The Netherlands

^b Delft University of Technology, Faculty of Civil Engineering and Geosciences, P.O. Box 5048, 2600 GA Delft, The Netherlands

^c Institute for Environmental Studies (IVM), Vrije Universiteit Amsterdam, 1081 HV Amsterdam, The Netherlands

^d *Instituto Universitario de Investigación Marina (INMAR) , University of Cádiz, 11510 Puerto Real, Cádiz, Spain*

^e *EO4 Data Science, 11500 El Puerto de Santa Maria, Cadiz, Spain*

References reply reviewer #3

- Athanasiou, Panagiotis, Ap van Dongeren, Alessio Giardino, Michalis I. Voudoukas, Roshanka Ranasinghe, and Jaap Kwadijk. 2020. "Uncertainties in Projections of Sandy Beach Erosion Due to Sea Level Rise: An Analysis at the European Scale." *Scientific Reports* 10(1).
- Brinkman, Richard Michael. 2006. "Wave Attenuation in Mangrove Forests: An Investigation through Field and Theoretical Studies." James Cook University.
- Cole, Thomas G., Katherine C. Ewel, and Nora N. Devoe. 1999. "Structure of Mangrove Trees and Forests in Micronesia." *Forest Ecology and Management* 117(1–3):95–109.
- Giri, C., E. Ochieng, L. L. Tieszen, Z. Zhu, A. Singh, T. Loveland, J. Masek, and N. Duke. 2011. "Status and Distribution of Mangrove Forests of the World Using Earth Observation Satellite Data." *Global Ecology and Biogeography* 20(1):154–59.
- Haer, Toon, W. J. Woute. Botzen, Vincent Van Roomen, Harry Connor, Jorge Zavala-Hidalgo, Dirk M. Eilander, and Philip J. Ward. 2018. "Coastal and River Flood Risk Analyses for Guiding Economically Optimal Flood Adaptation Policies: A Country-Scale Study for Mexico." *Philosophical Transactions of the Royal Society A: Mathematical, Physical and Engineering Sciences* 376(2121).
- Horstman, E. M., C. M. Dohmen-Janssen, P. M. F. Narra, N. J. F. van den Berg, M. Siemerink, and S. J. M. H. Hulscher. 2014. "Wave Attenuation in Mangroves: A Quantitative Approach to Field Observations." *Coastal Engineering* 94:47–62.
- Jadhav, Ranjit S., Qin Chen, and Jane M. Smith. 2013. "Spectral Distribution of Wave Energy Dissipation by Salt Marsh Vegetation." *Coastal Engineering* 77:99–107.
- Krauss, K. W., J. A. Allen, and D. R. Cahoon. 2003. "Differential Rates of Vertical Accretion and Elevation Change among Aerial Root Types in Micronesian Mangrove Forests." *Estuarine, Coastal and Shelf Science* 56(2):251–59.
- Mazda, Yoshihiro, Eric Wolanski, Brian King, Akira Sase, Daisuke Ohtsuka, and Michimasa Magi. 1997. "Drag Force Due to Vegetation in Mangrove Swamps." *Mangroves and Salt Marshes* 1(3):193–99.
- McGranahan, Gordon, Deborah Balk, and Bridget Anderson. 2007. "The Rising Tide: Assessing the Risks of Climate Change and Human Settlements in Low Elevation Coastal Zones." *Environment and Urbanization* 19(1):17–37.
- Mclvor, Anna, Iris Möller, Tom Spencer, and M. Spalding. 2012. *Reduction of Wind and Swell Waves by Mangroves*. Report 1: Cambridge Coastal Research Unit Working Paper 40. Cambridge.
- Mcowen, Chris, Lauren Weatherdon, Jan-Willem Bochove, Emma Sullivan, Simon Blyth, Christoph Zockler, Damon Stanwell-Smith, Naomi Kingston, Corinne Martin, Mark Spalding, and Steven Fletcher. 2017. "A Global Map of Saltmarshes." *Biodiversity Data Journal* 5:e11764.
- Menéndez, Pelayo, Iñigo J. Losada, Saul Torres-Ortega, Siddharth Narayan, and Michael W.

- Beck. 2020. "The Global Flood Protection Benefits of Mangroves." *Scientific Reports* 10(1):1–11.
- Möller, I. 2006. "Quantifying Saltmarsh Vegetation and Its Effect on Wave Height Dissipation : Results from a UK East Coast Saltmarsh." *Estuarine, Coastal and Shelf Science* 69:337–51.
- Möller, Iris, Matthias Kudella, Franziska Rupprecht, Tom Spencer, Maike Paul, Bregje K. Van Wesenbeeck, Guido Wolters, Kai Jensen, Tjeerd J. Bouma, Martin Miranda-lange, and Stefan Schimmels. 2014. "Wave Attenuation over Coastal Salt Marshes under Storm Surge Conditions." *Nature Geoscience* 7(September):727–32.
- Muis, Sanne, Martin Verlaan, Hessel C. Winsemius, Jeroen C. J. H. Aerts, and Philip J. Ward. 2016. "A Global Reanalysis of Storm Surges and Extreme Sea Levels." *Nature Communications* 7(May).
- Narayan, S. ..., T. .. Suzuki, M. J. Stive, H. J. Verhagen, W. Ursem, and R. Ranasinghe. 2011. "The Effectiveness of Mangroves in Attenuating Cyclone-Induced Waves." *Coastal Engineering Proceedings* 1.
- Rupprecht, F., I. Möller, M. Paul, M. Kudella, T. Spencer, B. K. van Wesenbeeck, G. Wolters, K. Jensen, T. J. Bouma, M. Miranda-Lange, and S. Schimmels. 2017. "Vegetation-Wave Interactions in Salt Marshes under Storm Surge Conditions." *Ecological Engineering* 100:301–15.
- Vousdoukas, Michalis I., Lorenzo Mentaschi, Jochen Hinkel, Philip J. Ward, Ignazio Mongelli, Juan-Carlos Ciscar, and Luc Feyen. 2020. "Economic Motivation for Raising Coastal Flood Defenses in Europe." *Nature Communications* 11(2119).
- Vousdoukas, Michalis I., Lorenzo Mentaschi, Evangelos Voukouvalas, Alessandra Bianchi, Francesco Dottori, and Luc Feyen. 2018. "Climatic and Socioeconomic Controls of Future Coastal Flood Risk in Europe." *Nature Climate Change* 8(9):776–80.
- Vuik, V., S. N. Jonkman, B. W. Borsje, and T. Suzuki. 2016. "Nature-Based Flood Protection: The Efficiency of Vegetated Foreshores for Reducing Wave Loads on Coastal Dikes." *Coastal Engineering* 116:42–56.
- Ward, Philip J., Brenden Jongman, Peter Salamon, Alanna Simpson, Paul Bates, Tom De Groeve, Sanne Muis, Erin Coughlan de Perez, Roberto Rudari, Mark A. Trigg, and Hessel C. Winsemius. 2015. "Usefulness and Limitations of Global Flood Risk Models." *Nature Climate Change* 5(8):712–15.
- Willemsen, Pim W. J. M., Bas W. Borsje, Vincent Vuik, Tjeerd J. Bouma, and Suzanne J. M. H. Hulscher. 2020. "Field-Based Decadal Wave Attenuating Capacity of Combined Tidal Flats and Salt Marshes." *Coastal Engineering* 156(December 2019):103628.
- Zhang, Xiaofeng, Vivien P. Chua, and Hin Fatt Cheong. 2015. "Hydrodynamics in Mangrove Prop Roots and Their Physical Properties." *Journal of Hydro-Environment Research* 9(2):281–94.

Reviewers' Comments:

Reviewer #1:

Remarks to the Author:

Dear authors,

I reviewed the your thoroughly revised manuscript and I appreciate the extensive effort to improve the results and text. I think the paper is now pretty much ready for publication, as the results are now more meaningful and robust, the methods were clarified and figures improved considerably. All questions were answered in full and acted on were necessary. I only have comments regarding your responses to three of my questions.

My concern remains regarding questions 9 and 10. I'm putting much emphasis on the economic computation partly because there is a lack of clarity on the calculation chain in the authors' descriptions (even in the revision), partly due to the impact on the results (as noticeable from Suppl. Fig. 10) and partly because I noticed a propensity (even in Nature publications) to incorrectly compute the transformation of prices across countries and time. I think a numerical example in the supplement would be useful to show the accuracy of the calculation. For greater transparency, rather than using a commercial, not accessible publication, I suggest taking World Bank's 2005 PPPs for construction costs. In this way, your generic assumption of levee cost of 7 million USD in the USA, where costs are 166.6% of world average, can be transformed into a construction cost specific for each country. In this way, your global estimate of protection benefit will be in world-averaged prices and the costs as % of GDP will take into account the fact that PPPs for construction are different from those for GDP. The data is accessible from World Bank's DataBank as part of "International Comparison Program (ICP) 2005" dataset. I have attached the data extracted from this database to this review.

Regarding question 8, not really a comment, just a suggestion: in the caption of Fig. 2 it would be good to mention that the construction costs are the "medium" scenario. In the abstract, it might be good to add the uncertainty bounds to your headline figures, based on the uncertainty of the construction costs (3x more, 3x less). Also, probably this will be fixed in the typesetting, but not to forget: the "2005" next to "USD" should be in a subscript.

I look forward to your final version of this important manuscript and (I'm confident) the published paper!

Best regards

Dominik Paprotny

Reviewer #2:

Remarks to the Author:

van Zeist and colleagues have taken considerably care in revising their paper and this has improved the manuscript in several major ways. For example they now include 1) a new figure 3 and the results of the sensitivity analysis, 2) the discussion of the replacement cost approach and avoided damages and the revised analysis of coastal protection needed, and 3) by placing the paper in the broader context of the broader ecosystem services literature. However, despite this improvement, I still have reservations about its suitability for Nat Climate Change.

1) The clarity of the writing is still poor and not suitable for such a high impact journal. I suggest the editor either rejects the paper at this point or the authors enlist an editor (outside the author team) whose first language is English and who writes high impact papers to help them go through each sentence and paragraph to ensure it is readable and polished. The paper simply cannot be published with the colloquialisms, the confusing sentences, the missing words, and the jargon.

Here are just a couple of examples of the lines with problems. But I'm not going to go through and fix the whole paper – that isn't my job as the reviewer: line 25 missing "are", the abstract says "role" three times, "delta" is in the first line of the abstract but the paper doesn't just focus on deltas, lines 195-196 incorrect sentence constructions, lines 218-219 confusing, lines 150-170 need edits throughout, the discussion needs to be further polished. And this is just a small selection of the problems.

2) Similarly, a few critical sentences for interpretation lack clarity e.g., line 251 – undercuts all that we do know about vegetation playing an important role of vegetation; lines 205-206 – undercuts the importance of global analysis.

3) I'm worried that this paper is now less of a major advance than when the authors first submitted it and I first reviewed it. A couple of key papers have come out. The first is Menendez et al. 2020 in Science Reports that models mangrove contributions to coastal risk reduction globally. This paper does include other wetland and saltmarsh habitat which is very important and the authors address this. The editor will have to decide if this is substantial enough of an advance for NCC. Second, the analysis does not include any spatial variation in vegetation characteristics. In 2019 when I first reviewed this paper, this was a pretty typical approach, but increasingly spatial data are available to account for variation in key vegetation attributes that influence wave attenuation (e.g., Simard et al. 2019 and subsequent papers which provide Mangrove canopy height globally as it relates to precipitation, temperature, and cyclone frequency).

At a minimum, the authors should mention briefly the assumption that all vegetation attributes are spatially constant in the discussion and not just in the methods and flag this as an area for future work – to combine their approach with new information EO data.

4) I'm glad the authors have now included the discussion of SLR in the paragraph beginning on line 254. This is important. However, this paragraph also made me realize that this paper actually doesn't address climate impacts. If I understand correctly, the analysis assumes current conditions. Thus, I'm not sure NCC is the appropriate journal.

5) Need a clear definition of coastal protection need / susceptible coastlines at the end of the introduction along with the other broad discussion of the approach. It takes the reader by surprise to see it on line 117 and then to have to go into the methods.

6) I think there are now way too many references for the limits of NCC.

7) Based on the sensitivity analysis the abstract should include a range of values of cost reduction.

Reviewer #3:

Remarks to the Author:

This paper has been much improved in light of the reviewer comments. Upon reading the authors' responses and the revised sections of the manuscript, I am convinced that the authors have adequately addressed the comments bar a few remaining queries / requested changes I include below. The authors have significantly revised the manuscript and addressed the key points of the reviewer around sensitivity testing and missing details on certain methods and assumptions. The manuscript has been much strengthened and benefits from the extra clarity around assumptions and around the potential for translating global to local interpretations, which had been the main concerns. On the basis of the revisions and the assumption that the authors can address the small remaining issues listed below, I recommend minor revisions.

Greatly welcome the introduction of Fig. 3 and Supplementary Fig. 10 and the information provided therein giving insights into the sensitivity of the cost computation to the various input parameters. It seems the authors have somewhat misunderstood Question 4 of Reviewer 3. The point being made here was precisely that the discussion of vegetation-induced wave dissipation needs to be made alongside a mention of the fact that, in addition to introducing vegetation at the bed, coastal ecosystems are in fact bio-engineers and create the intertidal profile themselves, with shallow water resulting from their growth as long as sufficient sediment supply is available (see also ref. 52 in the manuscript). Please add a statement to this effect in the introductory section.

Response to Reviewer 3, Question 36: could the authors clarify where in the revised manuscript this change was made.

Line 160: It is important here to state that the decrease in vegetation width with increase in population density is not a statistically significant trend (unless the authors can add a statistical statement / methods to show that the trend is statistically significant at an appropriate confidence level). Please add `...although this trend is not statistically significant given the large error margins`. I did not proof read the manuscript but noticed a few typos (see below) so would recommend full proof reading.

Line 281: typo `topograhly`

Line 287: `hinter` should be `hinder`

Line 307: avoid sensationalist language – replace `enormous` with `considerable`.

Line 310: as above – replace `massive` with `widespread` or `large degree of`

Point-by-point reply

revision Cutting the costs of coastal protection

Date

24 July 2021

Contact person

Vincent van Zelst

Number of pages

1 of 11

E-mail

Vincent.vanZelst@deltares.nl

Reviewer #1

General comments:

Reviewer #1 (Remarks to the Author):

Dear authors,

I reviewed the your thoroughly revised manuscript and I appreciate the extensive effort to improve the results and text. I think the paper is now pretty much ready for publication, as the results are now more meaningful and robust, the methods were clarified and figures improved considerably. All questions were answered in full and acted on were necessary. I only have comments regarding your responses to three of my questions.

I look forward to your final version of this important manuscript and (I'm confident) the published paper!

Best regards
Dominik Paprotny

Reply:

We are happy to read that the revised manuscript and the provided answers have been well received by reviewer #1. We like to thank reviewer #1 again for his efforts. Below we provided a response to the remaining comments.

Comment 1:

My concern remains regarding questions 9 and 10. I'm putting much emphasis on the economic computation partly because there is a lack of clarity on the calculation chain in the authors' descriptions (even in the revision), partly due to the impact on the results (as noticeable from Suppl. Fig. 10) and partly because I noticed a propensity (even in Nature publications) to incorrectly compute the transformation of prices across countries and time. I think a numerical example in the supplement would be useful to show the accuracy of the calculation.

For greater transparency, rather than using a commercial, not accessible publication, I suggest taking World Bank's 2005 PPPs for construction costs. In this way, your generic assumption of levee cost of 7 million USD in the USA, where costs are 166.6% of world average, can be transformed into a construction cost specific for each country. In this way, your global estimate of protection benefit will be in world-averaged prices and the costs as % of GDP will take into account the fact that PPPs for construction are different from those for GDP. The data is accessible from World Bank's DataBank as part of "International Comparison Program (ICP) 2005" dataset. I have attached the data extracted from this database to this review.

Reply:

We agree that this is an important point of concern. During evaluation of the code we noticed incorrect conversion from USD at market exchange rate to purchasing power parity (PPP). We updated the revised manuscript after correction. The main messages and implications of the paper remain unchanged. We inserted the formulation and a numerical example in the Methods (revised manuscript l. 183 – 184).

$$\text{Unit levee cost}_{\text{country}_i} = \text{unit levee cost} \cdot \text{construction index}_{\text{country}_i} / \text{PPP MER rate}_{2005} \text{ index}_{\text{country}_i}$$

Using this approach, we have included both the effect of global differences in construction costs (Figure 1 top) and in purchasing power (Figure 1 middle). This results in levee construction costs specific for each country (Figure 1 bottom). I discussed the suggestion of using the World Bank’s 2005 PPPs for construction costs within the authorship. We really prefer to keep onto the followed approach. Firstly, to be in line with previous publications that used this method (e.g. Ward et al. (2010) and Ward et al. (2017)). Secondly, because the currently applied construction index factors apply specifically to *civil engineering* construction works. For the reviewer’s information, we included a table of the applied indexes in Appendix A.

Figure 1 Global overview of levee unit cost. (Top) Construction index based on civil engineering construction costs, (Middle) PPP Market Exchange Rate 2005 index, (Bottom) Levee unit cost per country (mid scenario).

Comment 3:

Regarding question 8, not really a comment, just a suggestion: in the caption of Fig. 2 it would be good to mention that the construction costs are the “medium” scenario. In the abstract, it might be good to add the uncertainty bounds to your headline figures, based on the uncertainty of the construction costs (3x more, 3x less). Also, probably this will be fixed in the typesetting, but not to forget: the “2005” next to “USD” should be in a subscript.

Reply:

Following the suggestion of reviewer #1 we changed the caption of Fig. 2 in the revised manuscript and inserted ranges to the headline figures based on the uncertainty of the construction costs. In addition, we changed “2005” to subscript where applicable.

Reviewer #2

General comments:

Reviewer #2 (Remarks to the Author):

van Zelst and colleagues have taken considerably care in revising their paper and this has improved the manuscript in several major ways. For example they now include:

- 1) a new figure 3 and the results of the sensitivity analysis,
- 2) the discussion of the replacement cost approach and avoided damages and the revised analysis of coastal protection needed, and
- 3) by placing the paper in the broader context of the broader ecosystem services literature.

However, despite this improvement, I still have reservations about its suitability for Nat Climate Change.

Reply:

We like to thank Reviewer 2 for his/her efforts to review the revised manuscript. After reading the second revision, we sincerely hope that the Reviewer 2 has no more reservations about the suitability of the manuscript in *Nature Communications* (note: not *Nature Climate Change*).

Comment 1:

The clarity of the writing is still poor and not suitable for such a high impact journal. I suggest the editor either rejects the paper at this point or the authors enlist an editor (outside the author team) whose first language is English and who writes high impact papers to help them go through each sentence and paragraph to ensure it is readable and polished. The paper simply cannot be published with the colloquialisms, the confusing sentences, the missing words, and the jargon.

Here are just a couple of examples of the lines with problems. But I'm not going to go through and fix the whole paper – that isn't my job as the reviewer: line 25 missing "are", the abstract says "role" three times, "delta" is in the first line of the abstract but the paper doesn't just focus on deltas, lines 195-196 incorrect sentence constructions, lines 218-219 confusing, lines 150-170 need edits throughout, the discussion needs to be further polished. And this is just a small selection of the problems.

Reply:

The second revision of the manuscript has been checked and thoroughly re-edit by one of the co-authors who is a native English speaker and has published several high impact papers. We attached a version of the revised manuscript with track changes.

Comment 2:

Similarly, a few critical sentences for interpretation lack clarity e.g., line 251 – undercuts all that we do know about vegetation playing an important role of vegetation; lines 205-206 – undercuts the importance of global analysis.

Reply:

We rewrote the highlighted sentences to improve clarity. Revised manuscript lines 260 - 267 and 225 - 227.

Comment 3:

I'm worried that this paper is now less of a major advance than when the authors first submitted it and I first reviewed it. A couple of key papers have come out. The first is Menendez et al. 2020 in Science Reports that models mangrove contributions to coastal risk reduction globally. This paper does include other wetland and saltmarsh habitat which is very important and the authors address this. The editor will have to decide if this is substantial enough of an advance for NCC. Second, the analysis does not include any spatial variation in vegetation characteristics. In 2019 when I first reviewed this paper, this was a pretty typical approach, but increasingly spatial data are available to account for variation in key vegetation attributes that influence wave attenuation (e.g., Simard et al. 2019 and subsequent papers which provide Mangrove canopy height globally as it relates to precipitation, temperature, and cyclone frequency). At a minimum, the authors should mention briefly the assumption that all vegetation attributes are spatially constant in the discussion and not just in the methods and flag this as an area for future work – to combine their approach with new information EO data.

Reply:

In the revised manuscript we inserted text on the used key vegetation attributes and the opportunity to in cooperate spatially varying vegetation characteristics in future work (lines 254 – 260). Here, we also inserted reference to Simard et al. (2019).

Aspects that make the current study novel are indeed the inclusion of both mangroves and salt marshes in a process-based approach. In addition, as mentioned also during the first revision round, what makes the manuscript unique is that it bridges two fields (flood risk and ecosystem services), because the manuscript focusses on hybrid coastal protection (combination of vegetated foreshore with a levee). The vegetated foreshore attenuates incoming waves and the levee defenses against high waters. This design uses the strength of both modules. Mangroves and salt marshes are often capable of attenuating incoming waves, but fail to prevent flooding as high water (surges) are hardly affected in absence of very extensive wetland's. At the other hand, a purely 'grey' solution (a traditional levee or flood wall) is very costly if it has to withstand incoming waves at full force, but becomes financially attractive if combined with wetlands that attenuate these incoming waves. In addition, these hybrid solutions have the same co-benefits as full 'green' infrastructure, e.g. carbon storage, sediment trapping.

Comment 4:

I'm glad the authors have now included the discussion of SLR in the paragraph beginning on line 254. This is important. However, this paragraph also made me realize that this paper actually doesn't address climate impacts. If I understand correctly, the analysis assumes current conditions. Thus, I'm not sure NCC is the appropriate journal.

Reply:

The analysis indeed assumes current conditions. We agree that *Nature Climate Change* might not be the best fit. However, we suspect that the reviewer might be confused, because we are in the process for publication in *Nature Communications*. A journal that to the authorship seems very appropriate for this work.

Comment 5:

Need a clear definition of coastal protection need / susceptible coastlines at the end of the introduction along with the other broad discussion of the approach. It takes the reader by surprise to see it on line 117 and then to have to go into the methods.

Reply:

In the revised manuscript we inserted in the introduction that populated coastlines susceptible to flooding have been identified using flood maps of 1 km resolution and population density classes (lines 87-89). A detailed description of the derivation procedure can be found in the Methods, because this is too detailed to insert along the broad description of the approach in the introduction. In addition, the text has been re-edit throughout to improve clarity on this matter (e.g. lines 111-112).

Comment 6:

I think there are now way too many references for the limits of NCC.

Reply:

Reviewer #2 refers again to *Nature Climate Change*. However, in the submission guide for *Nature Communications* we read a guidance to 70 references. Currently the manuscript contains 94 references. I have seen *Nature Communication* Articles with over 100 references. However, we are open to have another critical look at the references if the editor prefers to lower the amount of citations in the revised manuscript.

Comment 7:

Based on the sensitivity analysis the abstract should include a range of values of cost reduction.

Reply:

This comment of Reviewer #2 is in line with the comment of Reviewer #1. We have inserted ranges to the headline figures.

Reviewer #3

Reviewer #3 (Remarks to the Author):

General comments:

This paper has been much improved in light of the reviewer comments. Upon reading the authors' responses and the revised sections of the manuscript, I am convinced that the authors have adequately addressed the comments bar a few remaining queries / requested changes I include below.

The authors have significantly revised the manuscript and addressed the key points of the reviewer around sensitivity testing and missing details on certain methods and assumptions. The manuscript has been much strengthened and benefits from the extra clarity around assumptions and around the potential for translating global to local interpretations, which had been the main concerns. On the basis of the revisions and the assumption that the authors can address the small remaining issues listed below, I recommend *minor revisions*. Greatly welcome the introduction of Fig. 3 and Supplementary Fig. 10 and the information provided therein giving insights into the sensitivity of the cost computation to the various input parameters.

Reply:

We are very happy that the first revision has been well received by Reviewer #3. We like to thank Reviewer #3 again for his/her time and efforts for reviewing the revision. We hope that after addressing the remaining issues the manuscript is suitable for publication.

Comment 1:

It seems the authors have somewhat misunderstood Question 4 of Reviewer 3. The point being made here was precisely that the discussion of vegetation-induced wave dissipation needs to be made alongside a mention of the fact that, in addition to introducing vegetation at the bed, coastal ecosystems are in fact bio-engineers and create the intertidal profile themselves, with shallow water resulting from their growth as long as sufficient sediment supply is available (see also ref. 52 in the manuscript). Please add a statement to this effect in the introductory section.

Reply:

We do agree with the reviewer that it is important to highlight this effect (early) in the manuscript. In the revised manuscript we inserted extra sentences (lines 59-64). Here, we also underline that the current study focusses on the present situation.

Comment 2:

Response to Reviewer 3, Question 36: could the authors clarify where in the revised manuscript this change was made.

Reply:

Question 36 of Reviewer #3 in the first revision round targeted the vegetation characteristics (specifically for mangrove vegetation) and the missing literature. In the revised manuscript the reviewer can find the adaptations on lines 115-119 Methods. In addition, we inserted additional sentences on the used vegetation schematization on lines 254-260 in the Discussion.

Comment 3:

Line 160: It is important here to state that the decrease in vegetation width with increase in population density is not a statistically significant trend (unless the authors can add a statistical statement / methods to show that the trend is statistically significant at an appropriate confidence level). Please add '...although this trend is not statistically significant given the large error margins'.

Reply:

The suggested sentence has been inserted on lines 169-170.

Comment 4:

I did not proof read the manuscript but noticed a few typos (see below) so would recommend full proof reading.

Line 281: typo 'topograh'y'

Line 287: 'hinter' should be 'hinder'

Line 307: avoid sensationalist language – replace 'enormous' with 'considerable'.

Line 310: as above – replace 'massive' with 'widespread' or 'large degree of'

Reply:

The second revision of the manuscript has been checked and thoroughly re-edit by one of the co-authors who is a native English speaker and has published several high impact papers. We attached a version of the revised manuscript with track changes.

A Appendix: Levee unit costs per country

Table 1 Levee unit costs per country. FID_Aqua = unique country ID in FAST Aqueduct project, ISO = ISO Alpha-3 country codes, Con_index = construction index based on civil engineering construction costs, PPP_index = PPP_Market Exchange Rate_2005_index, Levee unit cost = levee unit cost per country in mln USD₂₀₀₅ PPP km m⁻¹. Please do not disclose.

FID_Aque	ISO	NAME_ENGLI	Con_index	PPP_index	Levee unit cost
3	ALB	Albania	0.950	0.425	15.6
7	AGO	Angola	0.970	0.470	14.4
11	ARG	Argentina	0.920	0.437	14.7
14	AUS	Australia	1.000	1.059	6.6
17	BHS	Bahamas	0.991	0.917	7.6
19	BGD	Bangladesh	0.920	0.373	17.2
23	BLZ	Belize	0.970	0.611	11.1
32	BRA	Brazil	0.910	0.562	11.3
39	KHM	Cambodia	0.897	0.312	20.1
40	CMR	Cameroon	0.952	0.476	14.0
41	CAN	Canada	0.990	1.002	6.9
47	CHL	Chile	0.950	0.597	11.1
48	CHN	China	0.800	0.421	13.3
52	COL	Colombia	0.940	0.466	14.1
55	CRI	Costa Rica	0.937	0.512	12.8
57	HRV	Croatia	0.950	0.657	10.1
58	CUB	Cuba	0.937	0.522	12.6
63	DNK	Denmark	1.030	1.433	5.0
65	DOM	Dominican Republic	0.937	0.567	11.6
67	ECU	Ecuador	0.950	0.420	15.8
68	EGY	Egypt	0.850	0.269	22.1
69	SLV	El Salvador	0.937	0.496	13.2
70	GNQ	Equatorial Guinea	0.950	0.545	12.2
76	FJI	Fiji	0.897	0.846	7.4
77	FIN	Finland	1.100	1.218	6.3
78	FRA	France	1.000	1.151	6.1
79	GUF	French Guiana	0.937	0.438	15.0
82	GAB	Gabon	0.952	0.486	13.7
85	DEU	Germany	1.000	1.082	6.5
86	GHA	Ghana	0.940	0.666	9.9
88	GRC	Greece	0.960	0.898	7.5
93	GTM	Guatemala	0.937	0.527	12.4
95	GNB	Guinea-Bissau	0.952	0.413	16.1
96	GIN	Guinea	0.960	0.335	20.1
97	GUY	Guyana	0.937	0.693	9.5
98	HTI	Haiti	0.937	0.451	14.6
100	HND	Honduras	0.937	0.431	15.2

101	HKG	Hong Kong	0.991	0.731	9.5
103	ISL	Iceland	0.991	1.575	4.4
104	IND	India	0.840	0.333	17.7
105	IDN	Indonesia	0.850	0.406	14.7
106	IRN	Iran	0.940	0.292	22.5
108	IRL	Ireland	0.990	1.259	5.5
111	ITA	Italy	0.940	1.074	6.1
112	JAM	Jamaica	0.937	0.599	11.0
113	JPN	Japan	1.030	1.175	6.1
117	KEN	Kenya	0.970	0.391	17.4
118	KIR	Kiribati	0.897	0.438	14.3
126	LBR	Liberia	0.960	0.504	13.3
127	LBY	Libya	0.940	0.562	11.7
133	MDG	Madagascar	0.952	0.324	20.6
135	MYS	Malaysia	0.910	0.458	13.9
144	MEX	Mexico	0.880	0.654	9.4
151	MAR	Morocco	0.950	0.550	12.1
152	MOZ	Mozambique	0.952	0.473	14.1
153	MMR	Myanmar	0.897	0.252	24.9
157	NLD	Netherlands	1.020	1.116	6.4
158	NCL	New Caledonia	0.991	0.882	7.9
159	NZL	New Zealand	1.030	1.081	6.7
160	NIC	Nicaragua	0.937	0.385	17.0
162	NGA	Nigeria	0.940	0.459	14.3
165	PRK	North Korea	0.897	0.399	15.8
169	OMN	Oman	0.990	0.604	11.5
170	PAK	Pakistan	0.930	0.322	20.2
173	PAN	Panama	0.930	0.521	12.5
174	PNG	Papua New Guinea	0.897	0.428	14.7
177	PER	Peru	0.940	0.451	14.6
178	PHL	Philippines	0.930	0.395	16.5
181	PRT	Portugal	0.880	0.853	7.2
183	QAT	Qatar	0.991	0.780	8.9
187	RUS	Russia	0.991	0.450	15.4
199	SAU	Saudi Arabia	0.950	0.644	10.3
200	SEN	Senegal	0.952	0.479	13.9
203	SLE	Sierra Leone	0.952	0.364	18.3
204	SGP	Singapore	0.980	0.648	10.6
208	SLB	Solomon Islands	0.897	0.425	14.8
209	SOM	Somalia	0.952	0.399	16.7
210	ZAF	South Africa	0.930	0.609	10.7
212	KOR	South Korea	0.991	0.770	9.0
214	ESP	Spain	0.930	0.952	6.8
216	LKA	Sri Lanka	0.970	0.350	19.4

218	SUR	Suriname	0.937	0.586	11.2
221	SWE	Sweden	1.050	1.255	5.9
224	TWN	Taiwan	0.991	0.882	7.9
226	TZA	Tanzania	0.952	0.340	19.6
227	THA	Thailand	0.940	0.396	16.6
228	TLS	East Timor	0.897	0.438	14.3
233	TUN	Tunisia	0.940	0.499	13.2
234	TUR	Turkey	0.950	0.618	10.8
240	ARE	United Arab Emirates	0.980	0.902	7.6
241	GBR	United Kingdom	1.040	1.158	6.3
243	USA	United States	1.000	1.000	7.0
246	VUT	Vanuatu	0.897	0.515	12.2
248	VEN	Venezuela	0.960	0.546	12.3
249	VNM	Vietnam	0.897	0.297	21.1
253	YEM	Yemen	0.940	0.363	18.1

Reviewers' Comments:

Reviewer #1:

Remarks to the Author:

Dear authors,

I have no further comments. I'm looking forward to your final publication.

Best regards,

Dominik Paprotny

Reviewer #3:

Remarks to the Author:

I have read the authors' revised manuscript and their responses to reviewers' questions/concerns.

The authors have responded to the reviewer comments for the most part sufficiently to allow publication subject to clarification around some key remaining issues listed below.

Comment 1:

In addition to clarifications around text relating to my own previous comments, am now most concerned around the authors' response to Reviewer #2's comment 3, which is a very valid point. In their response, the authors seem to suggest that 'grey' solutions towards coastal flood risk can be set aside 'full 'green' infrastructure'. The authors also refer to the vegetated foreshore and the artificially designed levee as a 'modules'. The former suggests that coastal wetlands are considered as 'infrastructure'; the latter suggests that they can simply be used as an engineering 'module' alongside other such fixed infrastructural modules. This, to me, is rather revealing in the context of this paper, in which the main output is a financial argument for incorporating vegetated foreshores as a means by which to lower the cost of coastal protection (notwithstanding its ecosystem services benefits offered as an 'add on'), instead of seeing vegetated foreshores as the result of certain contextual conditions (geological setting, sediment supply, tidal and fluvial setting) in which they can thrive as biogeomorphological systems (i.e. they are not structures and are intricately connected to their surroundings, with process connections that often extend to many tens of kilometers).

Specifically:

Lines 14-18: The statements here in the abstract are actually very important in framing the paper but are confusing – and lead to the confusion that is also expressed in Reviewer #2's comment 3, who notes concern that 'this paper is now less of a major advance than when the authors first submitted it'. The reason for that confusion here in the abstract is their sentences here:

First (line 14-15), the authors state that 'the ability of ecosystems to contribute to reducing coastal flooding' is well known. Then (line 17), they say that here, they 'evaluate the ability of coastal vegetation [...] to reduce coastal hazard through wave reduction'. This seems rather contradictory – if this reduction is already well known, then why are they evaluating it again here?

It seems to me that the contribution of this paper is the costing element – i.e. we have now known for a while that (a) coastal ecosystems are not an ALTERNATIVES to hard structures, but that (b) they fulfil a wave reduction function and can thus be used ALONGSIDE hard structures. What is missing, and where this paper can rightly argue it's place, is the cost-saving that can be achieved by fully accounting for the presence of vegetated coastal ecosystems fronting such structures. THIS is what this paper is contributing and computing at a global scale, and thus this is the paper's novel

contribution.

This is clearer later (lines 49-52) but I would urge the authors to make the above much clearer in the abstract.

Comment 2:

Line 29-30: Given my comment above and Reviewer #2's points in Comment 3, this sentence also needs to be revised. It is not true that coastal areas are traditionally protected using human-made structures. Often, coastal areas are protected by natural features – the coast is in and of itself an energy buffer that protects the hinterland. It is only when people put themselves in areas that are not (sufficiently, for them) protected by the natural features present, that they end up being at risk... It is very important that this point is communicated.

Line 42: By the same token of the above, I would suggest to incorporate 'and large-scale relocation programmes are unlikely to be economically viable for the time being' behind 'developments' on Line 42.

Comment 3:

Line 64: I thank the authors for addressing my Comment 1 and adding the additional text in lines 59-64. However, the last sentence there needs to mention both wave and storm surge propagation being affected by decadal scale landform growth or decay (as the presence of the coastal wetland at a given elevation provides a structure around/over which water flows will be mediated, whether they be waves or mass movement of water during a storm surge – see e.g. the examples below:

Spencer, T., Brooks, S. M., Evans, B. R., Tempest, J. A., & Möller, I. (2015). Southern North Sea storm surge event of 5 December 2013: Water levels, waves and coastal impacts. *Earth-Science Reviews*, 146(December 2013), 120–145. <https://doi.org/10.1016/j.earscirev.2015.04.002>

Loder, N. M., Irish, J. L., Cialone, M. a. A., & Wamsley, T. V. (2009). Sensitivity of hurricane surge to morphological parameters of coastal wetlands. *Estuarine, Coastal and Shelf Science*, 84(4), 625–636. <https://doi.org/10.1016/j.ecss.2009.07.036>

Comment 4:

On line 249, the authors suggest that storm surges can only be reduced on vegetated foreshores of multiple kilometers. The Spencer et al. (2015) paper above appears to disprove that this is the case. The effect of coastal vegetation on storm surges is not merely a question of their width, as suggested by the authors in this paragraph – but also of the geomorphological nearshore configuration of the coast on which the wetlands are found. I suggest the authors modify the text in this paragraph to reflect that the relative impact of vegetated foreshores on storm surge water levels on any given coastline is highly context dependent.

Comment 5:

I am slightly confused by the revision of the text that is now on line 265 of the revised manuscript, where the authors suggest that further large-scale flume experiments are needed. When looking at the methods, I notice that the authors were using evidence from a large-scale flume experiment (Möller et al.) – could the authors be more specific here as to what kind of large-scale flume experiment would now be needed to advance the method?

Line-by-Line lacks of clarity

I appreciate that the authors have proof-read the manuscript, but there are still some vague statements / unclear wordings in there:

Line 67: 'More accurate...' – more accurate than what? Unclear. How accurate?

Line 87: two 'no' in here.

Line 75: presumably this should be 'alongshore distances'?

Line 124: full-stop missing

Line 288: 'Assessing the effects of...' – on what? Unclear. The statement prior to this is also unclear. 'All studies indicate....' is not an accurate reflection of the literature. There may be a theoretical threshold, but the 'critical threshold' is certainly extremely context (and most definitely sediment deliver) dependent.

Supplementary Fig 10: is 'mln' the right abbreviation for 'million'? 'M' is more common, I think.

Point-by-point reply - revision Cutting the costs of coastal protection.

Date

8 October 2021

Contact person

Vincent van Zelst

Number of pages

1 of 6

E-mail

Vincent.vanZelst@deltares.nl

Reviewer #1

General comments:

Reviewer #1 (Remarks to the Author):

Dear authors,

I have no further comments. I'm looking forward to your final publication.

Best regards,

Dominik Paprotny

Reply:

We are happy to read that the second revised manuscript and the provided answers have been well received by Reviewer #1. We like to thank Reviewer #1 again for his efforts.

Reviewer #3

Reviewer #3 (Remarks to the Author):

General comments:

I have read the authors' revised manuscript and their responses to reviewers' questions/concerns.

The authors have responded to the reviewer comments for the most part sufficiently to allow publication subject to clarification around some key remaining issues listed below.

Reply:

We are very happy that the second revision has been well received by Reviewer #3. We like to thank Reviewer #3 again for his/her time and efforts for reviewing the revision. We sincerely hope that after addressing the key remaining issues, Reviewer #3 could mark the revised manuscript suitable for publication.

Comment 1:

In addition to clarifications around text relating to my own previous comments, am now most concerned around the authors' response to Reviewer #2's comment 3, which is a very valid point. In their response, the authors seem to suggest that 'grey' solutions towards coastal flood risk can be set aside 'full 'green' infrastructure'. The authors also refer to the vegetated foreshore and the artificially designed levee as a 'modules'. The former suggests that coastal wetlands are considered as 'infrastructure'; the latter suggests that they can simply be used as an engineering 'module' alongside other such fixed infrastructural modules. This, to me, is rather revealing in the context of this paper, in which the main output is a financial argument for incorporating vegetated foreshores as a means by which to lower the cost of coastal protection (notwithstanding its ecosystem services benefits offered as an 'add on'), instead of seeing vegetated foreshores as the result of certain contextual conditions (geological setting, sediment supply, tidal and fluvial setting) in which they can thrive as biogeomorphological systems (i.e. they are not structures and are intricately connected to their surroundings, with process connections that often extend to many tens of kilometers).

Specifically:

Lines 14-18: The statements here in the abstract are actually very important in framing the paper but are confusing – and lead to the confusion that is also expressed in Reviewer #2's comment 3, who notes concern that 'this paper is now less of a major advance than when the authors first submitted it'. The reason for that confusion here in the abstract is their sentences here:

First (line 14-15), the authors state that 'the ability of ecosystems to contribute to reducing coastal flooding' is well known. Then (line 17), they say that here, they 'evaluate the ability of coastal vegetation [...] to reduce coastal hazard through wave reduction'. This seems rather contradictory – if this reduction is already well known, then why are they evaluating it again here?

It seems to me that the contribution of this paper is the costing element – i.e. we have now known for a while that (a) coastal ecosystems are not an ALTERNATIVES to hard structures, but that (b) they fulfil a wave reduction function and can thus be used ALONGSIDE hard structures. What is missing, and where this paper can rightly argue it's place, is the cost-saving that can be achieved by fully accounting for the presence of vegetated coastal ecosystems fronting such structures. THIS is what this paper is contributing and computing at a global scale, and thus this is the paper's novel contribution.

This is clearer later (lines 49-52) but I would urge the authors to make the above much clearer in the abstract.

Reply:

We fully agree with the reviewer's statements. We have rewritten the abstract line that caused confusion (revised manuscript l. 18-21). In addition, we have tried to further improve clarity on this matter by:

1. Changing the title of the manuscript.
2. Highlighting that coastal ecosystems are not alternatives, but complements to engineered flood defences (l. 36).
3. Inserting a concluding sentence that accounting for vegetation presence fronting levees could be cost-effective (l. 57-58).
4. Underlining that vegetated foreshores are not structures, but part of the ecosystem (l. 312).

Comment 2:

Line 29-30: Given my comment above and Reviewer #2's points in Comment 3, this sentence also needs to be revised. It is not true that coastal areas are traditionally protected using human-made structures. Often, coastal areas are protected by natural features – the coast is in and of itself an energy buffer that protects the hinterland. It is only when people put themselves in areas that are not (sufficiently, for them) protected by the natural features present, that they end up being at risk... It is very important that this point is communicated.

Line 42: By the same token of the above, I would suggest to incorporate 'and large-scale

relocation programmes are unlikely to be economically viable for the time being' behind 'developments' on Line 42.

Reply:

We inserted clarification on coastal protection by natural features and human-made structures (l. 32-34). We inserted that originally coastlines are protected by natural features, but coastal lines are often protected with human-made structures where occupied land is low lying or where people have put themselves coastwards.

As we did not look at the feasibility of large-scale relocation programmes, we prefer not to elaborate on relocation along lines 47-48.

Comment 3:

Line 64: I thank the authors for addressing my Comment 1 and adding the additional text in lines 59-64. However, the last sentence there needs to mention both wave and storm surge propagation being affected by decadal scale landform growth or decay (as the presence of the coastal wetland at a given elevation provides a structure around/over which water flows will be mediated, whether they be waves or mass movement of water during a storm surge – see e.g. the examples below:

Spencer, T., Brooks, S. M., Evans, B. R., Tempest, J. A., & Möller, I. (2015). Southern North Sea storm surge event of 5 December 2013: Water levels, waves and coastal impacts. *Earth-Science Reviews*, 146(December 2013), 120–145. <https://doi.org/10.1016/j.earscirev.2015.04.002>

Loder, N. M., Irish, J. L., Cialone, M. a. A., & Wamsley, T. V. (2009). Sensitivity of hurricane surge to morphological parameters of coastal wetlands. *Estuarine, Coastal and Shelf Science*, 84(4), 625–636. <https://doi.org/10.1016/j.ecss.2009.07.036>

Reply:

As the focus of the manuscript is on wave propagation, we initially omitted explicit naming of surges. We changed the highlighted lines according to the reviewer's suggestion (l. 71).

Comment 4:

On line 249, the authors suggest that storm surges can only be reduced on vegetated foreshores of multiple kilometers. The Spencer et al. (2015) paper above appears to disprove that this is the case. The effect of coastal vegetation on storm surges is not merely a question of their width, as suggested by the authors in this paragraph – but also of the geomorphological nearshore configuration of the coast on which the wetlands are found. I suggest the authors modify the text in this paragraph to reflect that the relative impact of

vegetated foreshores on storm surge water levels on any given coastline is highly context dependent.

Reply:

In the manuscript we highlighted that studies emphasized the importance of storm duration and storm intensity on surge reduction by vegetation. In the revised manuscript we inserted additional lines to further underline that there are contradicting studies on this topic, which suggests that the impact of vegetated foreshores on storm surges is context dependent (l. 258-260). Here, we inserted the study by Spencer et al. (2015) as additional example. Furthermore, in lines 308-310 we stressed that the global approach using coastal normal transects is insufficient to capture local deviations in topography.

Comment 5:

I am slightly confused by the revision of the text that is now on line 265 of the revised manuscript, where the authors suggest that further large-scale flume experiments are needed. When looking at the methods, I notice that the authors were using evidence from a large-scale flume experiment (Möller et al.) – could the authors be more specific here as to what kind of large-scale flume experiment would now be needed to advance the method?

Reply:

Apologies for the confusion. We indeed used evidence from a large-scale flume experiment (Möller et al., 2014) for salt marshes. However, we wanted to stress the absence of full-scale flume experiments with mangroves. We inserted clarification on this matter in lines 282-284 of the revised manuscript.

Comment 6:

I appreciate that the authors have proof-read the manuscript, but there are still some vague statements / unclear wordings in there.

Reply:

Line 67: 'More accurate...' – more accurate than what? Unclear. How accurate?

More accurate in comparison to available global elevation and bathymetry maps. We excluded the accuracy at this point in the manuscript to improve readability. In the revised manuscript we inserted the accuracy (l. 75-76).

Line 87: two 'no' in here.

Corrected in the revised manuscript.

Line 75: presumably this should be 'alongshore distances'?

Indeed. We clarified this in the revised manuscript (l. 83).

Line 124: full-stop missing

Corrected in the revised manuscript.

Line 288: 'Assessing the effects of....' – on what? Unclear. The statement prior to this is also unclear. 'All studies indicate....' is not an accurate reflection of the literature. There may be a theoretical threshold, but the 'critical threshold' is certainly extremely context (and most definitely sediment deliver) dependent.

We targeted here assessing the potential loss of coastal ecosystems globally due to SLR and coastal squeeze and the resulting effect of the partial loss of vegetation's wave buffering capacity on coastal protection costs. We changed the highlighted lines to improve clarity (l. 298-301).

Supplementary Fig 10: is 'mln' the right abbreviation for 'million'? 'M' is more common, I think. Mln is a rarer abbreviation to express million. 'M' is used as 'official' abbreviation in American English. 'm' is used as 'official' abbreviation in British English. To omit any confusion, we changed 'mln' to 'millions' in Supplementary Fig. 10.